# Photovoltaic power potential in West Africa using long-term satellite data

Ina Neher[1,2], Susanne Crewell[2], Stefanie Meilinger[1], Uwe Pfeifroth[3], and Jörg Trentmann[3]

[1]International Center for Sustainable Development, University of Applied Science Bonn-Rhein-Sieg, Grantham-Allee 20, 53757 Sankt Augustin, Germany
[2]Institute of Geophysics and Meteorology, University of Cologne, Albertus-Magnus-Platz, 50923 Köln, Germany
[3]Deutscher Wetterdienst, Satellite-based Climate Monitoring, Frankfurter Str. 135, 63067 Offenbach, Germany

**Correspondence:** Ina Neher (ina.neher@h-brs.de)

**Abstract.**

This paper addresses long-term historical changes in solar irradiance for West Africa ($3°$N to $20°$N and $20°$W to $16°$E) and its implications for photovoltaic systems. Here we use satellite irradiance (Surface Solar Radiation Data Set-Heliosat, Edition 2.1, SARAH-2.1) and temperature data from a reanalysis (ERA5) to derive photovoltaic yields. Based on 35 years of data (1983 - 2017) the temporal and regional variability as well as long-term trends of global and direct horizontal irradiance are analyzed. Furthermore, at four locations a detailed time series analysis is undertaken.

According to the high spatially resolved SARAH-2.1 data record ($0.05°$ x $0.05°$), solar irradiance is largest (with up to $300\,\mathrm{W/m^2}$ daily average) in the Sahara and the Sahel zone with a positive trend (up to 5 $(\mathrm{W/m^2})$/decade) and a lower temporal variability ($< 75\,\mathrm{W/m^2}$ between 1983 and 2017 for daily averages). Whereas, the solar irradiance is lower in southern West Africa (between $200\,\mathrm{W/m^2}$ and $250\,\mathrm{W/m^2}$) with a negative trend (up to -5 $(\mathrm{W/m^2})$/decade) and a higher temporal variability (up to $150\,\mathrm{W/m^2}$). The positive trend in the North is mostly connected to the dry season, while the negative trend in the South occurs during the wet season. Both trends show a 95% significance. PV yields show a strong meridional gradient with lowest values around $4\,\mathrm{kWh/kWp}$ in southern West Africa and reach more than $5.5\,\mathrm{kWh/kWp}$ in the Sahara and Sahel zone.

*Copyright statement.* TEXT

## 1 Introduction

The United Nations proposed the Sustainable Development Goals to achieve a better and more sustainable future (United Nations, 2015). The seventh goal, to "ensure access to affordable, reliable, sustainable and modern energy for all", implies a shift away from fossil-fuel based towards renewable energy sources. Especially for regions with high irradiance solar power is a promising option (e.g. Haegel et al., 2017; Solangi et al., 2011). However, potential sites and their yield need to be investigated carefully to ensure long-term sustainable investment.

With regard to energy availability and security West Africa is one of the least developed regions in the world (ECOWAS, 2017). Therefore, the power system will need to be strongly expanded in West Africa as there exists a gap between electricity supply and demand (Adeoye and Spataru, 2018). West Africa receives high amounts of global horizontal irradiance (GHI) (Solargis, 2019). Located within the descending branch of the Hadley Cell the Sahara and the Sahel zone are overall dry with little cloudiness leading to high sunshine duration (Kothe et al., 2017). Photovoltaic (PV) power seems to be a promising technology in this region. Thus, the development of a PV system is worthwhile. Before investing in a PV system three points need to be considered, using differently resolved global horizontal irradiance (GHI, the sum of direct (DIR) and diffuse horizontal irradiance (DHI)), which has the major impact on PV systems (Sengupta et al., 2017). First, to select a profitable location high spatially resolved GHI is needed. Second, to estimate the profitability and risks of the plant long-term variability and trends of historical GHI can be analyzed as a basis to project future system performance. And third, to optimize the plant high temporally resolved GHI can be used for the dimension of the plant size and storage system as well as for the maintenance. However, ground-based measurements of irradiance are not available continuously over long-term time scales and cover only a few discrete locations in the region.

Satellite based irradiance measurements have the advantage of being available for long time periods and covering wide spatial regions (Gueymard and Wilcox, 2011). Especially geostationary satellites can deliver data in a temporal resolution of less than one hour and a high spatial resolution. Therewith, potential PV yields can be calculated for selecting a profitable location as well as to analyze profitability and risks in a long-term view. Furthermore, such data sets enable the analysis of diurnal variability that needs to be taken into account for storage sizing and power system design. Here the first point is addressed, by using the daily averaged data to provide an overview on the potential PV yields over the full region. The European Organization for the Exploitation of Meteorological Satellites (EUMETSAT) Satellite Application Facility on Climate Monitoring (CM SAF) provides the Surface Solar Radiation Data Set-Heliosat, Edition 2.1 (SARAH-2.1), a 35 year long climate data record at half-hourly resolution, covering the whole of Africa and Europe (Pfeifroth et al., 2019a). The validation of this data set to stations from the Baseline Surface Radiation Network (BSRN) shows high quality, with a target accuracy of 15 $\text{W/m}^2$ (Pfeifroth et al., 2019b). However, only one of the BSRN stations lies close to the West African region (the majority of the stations are in Europe, see Pfeifroth et al. (2019b) for station details). As solar irradiance is affected by the atmosphere (cloud, aerosol and trace gases), several assumptions on optical properties need to be taken into account for the satellite data retrieval. Especially aerosol loads can be highly fluctuating (Neher et al., 2019; Slingo et al., 2006) and reach highest global values in the West African region (Kinne et al., 2013). Thus, a detailed validation of the full 35 year SARAH-2.1 data set for West Africa is needed and has not been performed so far.

Besides the atmospheric impact, solar irradiance reaching the top layer of a PV power module is affected by the solar zenith and the tilting angle of the module. Furthermore, soiling and reflections on the modules front and shade from the surrounding have an additional impact on the amount of radiation which can be transformed by the PV cell to a direct current. The cell temperature (impacted by the incoming irradiance, ambient temperature and wind speed) adjusts the efficiency of the PV cell (Skoplaki and Palyvos, 2009). Explicit models for PV power simulation are available (Neher et al., 2019; Ishaque et al., 2011;

King et al., 2004). However, they need explicit input data in a high temporal resolution which is often not available. Therefore, a simplified model for PV yield estimations based on daily data is developed and applied here.

In this study, the central research question "How do long-term atmospheric variability and trends impact photovoltaic yields in West Africa?" is answered by analyzing the SARAH-2.1 data record for West Africa. To give a comprehensive answer the article is structured along the following sub-questions.

- How accurate is the SARAH-2.1 data set for the considered region of West Africa?

- What are the trends and variability of solar irradiance between 1983 and 2017 in West Africa?

- How different are these trends and variability for varying latitudes and seasons?

- Which implications can be drawn for photovoltaic power?

This article is organized as follows. Section 2 introduces the ground and satellite based data. Methodologies to estimate photovoltaic power are described in Section 3. The satellite data validation with ground-based measurements is presented in Section 4. The variability and trend analysis of GHI and DIR for the time period from 1983 to 2017 is shown in Section 5. Furthermore, the temporal variability at different latitudes is analyzed. Section 6 estimates the implications of solar irradiance variability and trends for PV yields focusing on West Africa, using a simplified yield estimation based on measurements at three locations. Finally, the conclusions are given in Section 7.

## 2   Region overview and data sources

West Africa (in this study defined as the region from 3°N to 20°N and 20°W to 16°E) is a region with a pronounced dry and wet season. In large parts of West Africa one wet season occurs during the summer months. However, the length of the wet season decreases with rising latitude and along the coastal region, two wet seasons occur (typically in June/July and September). Nevertheless, here we use one single definition of seasons according to Mohr (2004) assuming one dry season: October - April and one wet season: May - September. To reinforce our results we performed the analysis with a sharper definition of seasons (dry: November - March and wet: June to August) and found similar results. The difference in seasons is mainly caused by the West African Monsoon (WAM) circulation and the Inter Tropical Convergence Zone (ITCZ). The ITCZ moves from north to south and back in an annual cycle according to the seasons (north during the wet and south during the dry season). West Africa is in general rather flat with highest elevations typically below 1000 m (Figure 1 a, Global Land One-km Base Elevation Project (GLOBE) database (Hastings and Dunbar, 1999)). Some exceptions are the Mount Cameroon on the south-east of the study area along the border of Nigeria and Cameroon, Fouta Djallon and the Guinea Highlands in Guinea, Jos Plateau in the center of Nigeria and the Aïr Mountains in northern Niger. Here, but locally also for lower mountain ranges, orographically enhanced cloudiness might occur. The enhanced cloudiness associated to the moist tropical region is clearly visible in the mean cloud albedo used as input for the SARAH-2.1 data retrieval between 1983 and 2017 (see Figure 1 b, from the SARHA-2.1 data set described later). Clouds have the major influence on the irradiance analyzed in this study. The West African climate

zones related to the albedo climatology (used for the SARAH-2.1 data retrieval), with a higher albedo of up to 0.35 in the desert region in the north and a lower albedo of down to 0.1 in the forest region in the south (see Figure 1 c, Surface and Atmospheric Radiation Budget (SARB) data from Clouds and the Earth's Radiant Energy System (CERES)). Frequent dust outbreaks occur over the total region (Cowie et al., 2014). Thereby, climatological highest aerosol optical depth (AOD) of up to 0.35 can be found in northern Mali (see Figure 1 d, from the European Center for Medium Range Weather Forecast, Monitoring Atmospheric Composition and Climate (MACC) and used for the SARAH-2.1 data retrieval). However, in local measurements AOD reach daily averages of up to 4 in the Sahel region (AERONET, 2014). Therewith, aerosols can have a high impact on the irradiance besides clouds and thus on solar power (Neher et al., 2019).

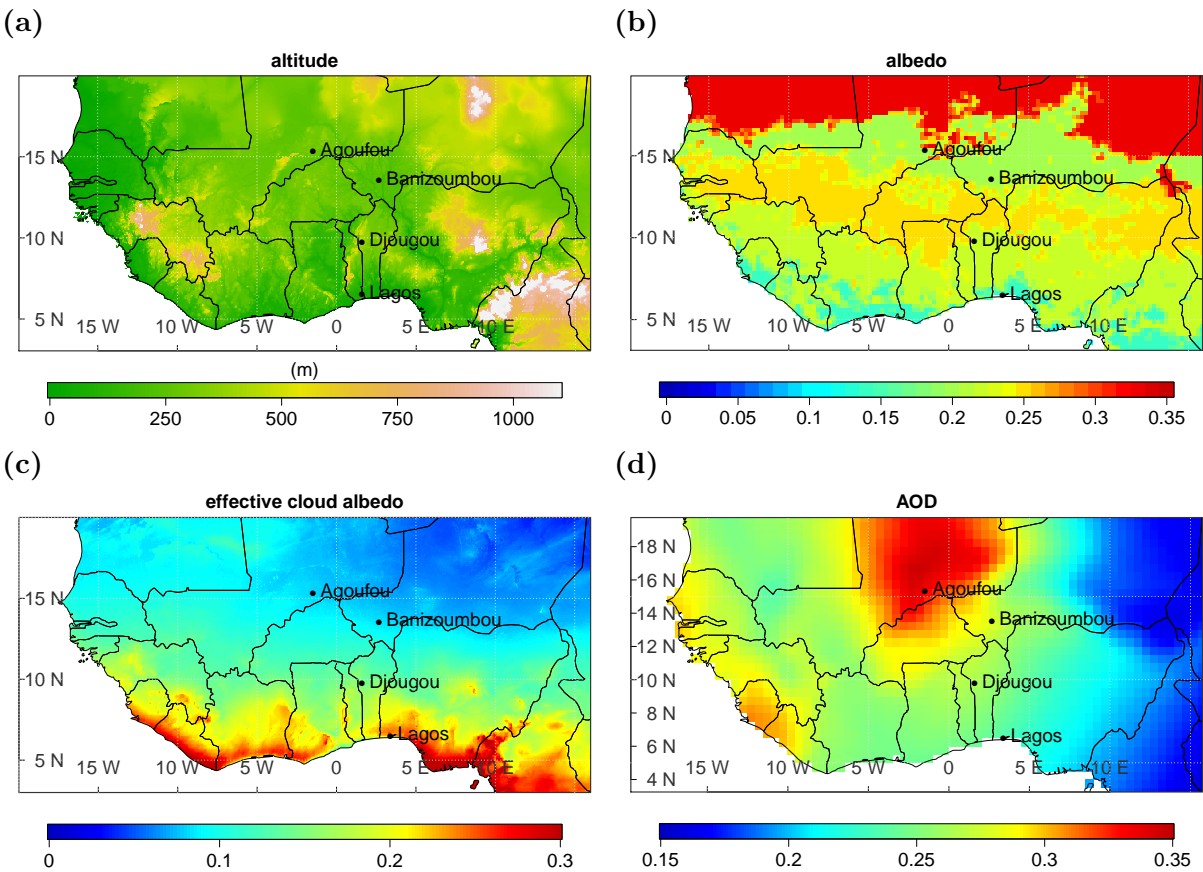

**Figure 1.** Topography of the considered region (a, Global Land One-km Base Elevation Project (GLOBE) database (Hastings and Dunbar, 1999)), mean cloud albedo between 1983 and 2017 (b, from the SARAH-2.1 data set described in Section 2.1), albedo climatology (c, Surface and Atmospheric Radiation Budget (SARB) data from Clouds and the Earth's Radiant Energy System (CERES)) and aerosol optical depth climatology (d, European Center for Medium Range Weather Forecast, Monitoring Atmospheric Composition and Climate (MACC)). Location of the three ground-based sites (Agoufou, Banizoumbou, Djougou) are marked as well as the additional location used for the time series analysis in Section 5.2 (Lagos).

## 2.1 Satellite-based data

The Surface Solar Radiation Data Record – Heliosat Edition 2.1 (SARAH-2.1) data set is provided by the EUMETSAT CM
SAF and covers the time period from 1983 to 2017 (Pfeifroth et al., 2019a, 2018). Besides others, the data set provides the
surface incoming shortwave radiation (GHI), the surface incoming direct radiation (DIR), the direct normal radiation (DNI)
and the effective cloud albedo (CAL). The products of SARAH-2.1 are retrieved from the geostationary METEOSAT satellite
service of the first and second generation, covering total West Africa with a half-hour temporal and a 0.05° x 0.05° spatial
resolution. For the retrieval, the Heliosat algorithm to estimate the effective cloud albedo (Hammer et al., 2003) is combined
with a cloud free radiative transfer model (Mueller et al., 2012). Furthermore, several climatological parameters are used for
the retrieval: the precipitable water vapor (ERA-interim), monthly AOD climatology (see Figure 1 d, MACC), monthly ozone
climatology (standard US atmosphere) and the surface albedo (see Figure 1 c, SARB data from CERES). For the generation
of the SARAH-2.1 data record the visible channel (0.5 - 0.9 μm) of the METEOSAT Visible and Infrared Imager (MVIRI) is
used until 2005 and the two visible channels (0.6 and 0.8 μm) of the Spinning Enhanced Visible and Infrared Imager (SEVIRI)
afterward. A detailed description of the retrieval is given in Mueller et al. (2015) and references within.

The advantages of the SARAH-2.1 data set compared to SARAH-1 are a higher stability in early years (due to the removal
of erroneous satellite images) and during the transition from the first to the second generation METEOSAT satellite in 2006.
Furthermore, the used water vapor climatology was topographically corrected and the consideration of situations with high
zenith angles were improved to account for an overestimation of cloud detection at low satellite viewing angles. A mean
absolute error (MAE, in comparison to 15 BSRN stations between 1994 and 2017) of 5.5 $W/m^2$ and 11.7 $W/m^2$ for monthly
and daily GHI is reached, respectively (Pfeifroth et al., 2019b).

In this study, the SARAH-2.1 data record (GHI and DIR in daily resolution) is used for the trend and variability analysis
over the whole 35 years and for the entire region. Daily and monthly means of GHI are compared to measured GHI at the
three AMMA sites. CMSAF SARAH-2.1 data is downloaded as daily and monthly averaged data. A detailed description of
the averaging approach can be found in Trentmann and Pfeifroth (2019). Instantaneous (half hourly) data is used to estimate
PV yields at the three AMMA sites to develop a simpler empirical PV model. The 30-min records were linearly interpolated
by using the diurnal cycle of the clear sky irradiance and the temporal resolution of the measured meteorological data (ambient
temperature and wind speed) was adjusted to the satellite data.

## 2.2 Ground-based data

Ground-based measurements of GHI complemented by ancillary data over several years are available from the African Mon-
soon Multidisciplinary Analysis (AMMA) program (AMMA, 2018; Redelsperger et al., 2006) at three sites (Agoufou, Mali;
Banizoumbou, Niger and Djougou, Benin, Figure 1). The sites are distributed over different land areas, one desert site, one site
in the Sahel region and one site in the Savanna. The data availability is limited to several years in the beginning of the $21^{st}$
century. All relevant parameters, including location, instrument information and measuring times are summarized in Table 1.

125 Additionally, measurements of ambient temperature and wind speed were taken during the AMMA campaign at the three sites. AMMA data is measured at a 15-minute resolution. To calculate robust daily averages, each of the 15 min values of a day needs to be available for calculating the mean. If only one measurement is missing, the day is disregarded. Monthly averages are calculated if there are at least 10 daily averages available over the month.

**Table 1.** Information on ground-based measuring sites.

| Station Name | Agoufou | Banizoumbou | Djougou |
|---|---|---|---|
| Country | Mali | Niger | Benin |
| Latitude | 15.3°N | 13.5°N | 9.7°N |
| Longitude | 1.5°W | 2.7°E | 1.6°E |
| Instrument | CNR1 | SKS 1110 | SP Lite2 |
| Accuracy | ±10% (daily totals) | ±5% | ±2.5% or 10W/m$^2$ |
| Reference | (Campbell Scientific, 2010) | (Skye Instruments, 2019) | (Kipp & Zonen, 2019) |
| Time | 2005 - 2011 | 2005 - 2012 | 2002 - 2009 |
| Resolution | 15 min | 15 min | 15 min |
| Land use | Desert | Sahel | Savanna |

At all sites, measurements of the aerosol robotic network (AERONET (Holben et al., 1998)) for the AOD are available. The
130 AOD measurements are retrieved from solar radiances at certain wavelengths and cloud screened in a post processing (Giles et al., 2019). Here, the quality assured data set of Level 2.0, Version 3 at 440 nm wavelength are used. For the comparison with daily satellite data (see Section 4) daily averages are downloaded from AERONET (AERONET, 2014) (thereby all data series of one day are averaged). Monthly averages are calculated as described before for the AMMA dataset.

## 3  Photovoltaic yield estimation

135 Our ultimate goal is to describe the PV potential over the entire region for a standardized PV power plant. For this purpose, a simplified linear regression is fitted on the basis of the three reference sites where the necessary information is available. Furthermore, the uncertainties concerning cell temperature are estimated (see Section 3.2) and the used GHI (from SARAH-2.1 data set) is validated (see Section 4). Therefore, ERA5 data is used (Hersbach et al., 2020; Copernicus Climate Change Service (C3S), 2017) for daily mean temperature. The ERA5 archive is based on a global reanalysis and is available from 1979
140 on. The single calculation steps, including all necessary input data is shown in Figure 2.

### 3.1  Model development

To estimate the PV power potential an empirical linear model is developed in a temporal resolution of one day, using a normalized GHI (from SARAH-2.1) as an input. This linear model is derived by simplifying the widely known two-diode-model (e.g. Ishaque et al., 2011). For this purpose, explicit PV power calculations are integrated over the diurnal cycle using

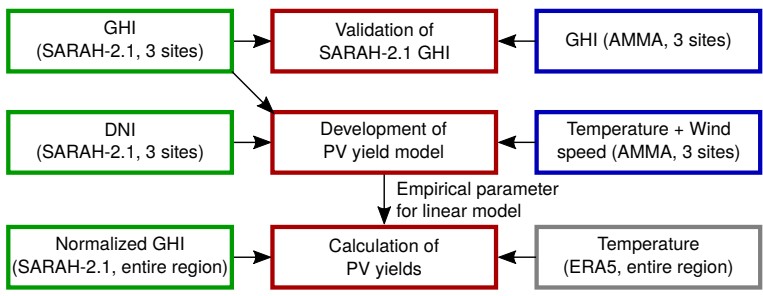

**Figure 2.** Connection of calculation steps (red) within this study, including all needed input data (green: satellite data, gray: reanalysis data, blue: observational data).

AMMA measurements (ambient temperature and wind speed) and SARAH-2.1 data (GHI and DIR) at the three measuring sites as input for the full model, serving as a reference and to train the linear model.

The two-diode-equation calculates the current ($I$) - voltage ($U$) - characteristics of a PV module from cell temperature $T_c$, global tilted irradiance (GTI) and typical modules characteristics

$$I(U) \quad = \quad I_{PH}(\text{GTI}, T_c) - I_{D1}(T_c) \left( e^{\frac{U + I \cdot R_S}{n_1 \cdot U_T}} - 1 \right) - \tag{1}$$

$$I_{D2}(T_c) \left( e^{\frac{U + I \cdot R_S}{n_2 \cdot U_T}} - 1 \right) - \frac{U + I \cdot R_S}{R_P}.$$

Thereby, two diodes ($D_1$ and $D_2$) are assumed in parallel, with differing saturation currents ($I_{D1}(T_c)$ and $I_{D2}(T_c)$), each depending on the cell temperature. The diode ideality factors ($n_1$ and $n_2$) are constant, with $n_1$=1 and $n_2$=2 (Salam et al., 2010). Furthermore, two resistors are connected, one in parallel ($R_P$) for the description of leakage currents and one in series ($R_S$) for the description of voltage drops, with a constant value for the system. The thermal voltage $U_T$ is proportional to the cell temperature. For the rayed solar cell a parallel current source can be assumed. The current source produces the photocurrent $I_{PH}(\text{GTI}, T_c)$ depending on the incoming solar irradiance and the cell temperature. Thus a simplification can be written as

$$I(U) \ = \ I_{PH}(\text{GTI}, T_c) + f(T_c, I, U), \tag{2}$$

with $f$ being a cell temperature, current and voltage dependent function.

The photocurrent depends linearly on the incoming tilted irradiance and is the major term of $I(U)$,

$$I_{PH} \ = \ \left( I_{SC}^{STC} + K_i(T_c - T_{STC}) \right) \frac{\text{GTI}}{GHI_{STC}}. \tag{3}$$

By assuming a typical silicon PV module (Solar world 235 poly (SolarWorld, 2012)), the modules characteristics are given with $I_{SC}^{STC} = 8.35$ A denoting the short circuit current at standard test conditions (STC), $K_i = 0.00034 \ I_{SC}/K$ being the temperature coefficient for the current, $T_{STC} = 25°C$ and $GHI_{STC} = 1000 \ \text{W/m}^2$ being the STC conditions for PV modules. By simplifying Equation 3 with $I_{SC}^{STC} \gg K_i(T_c - T_{STC})$ (for the typical cell temperature of $46°C$, used in the PV community and STC the right term would be 0.06 A), the temperature dependence is ignored here. The maximum-power-point (MPP) is

calculated as the product of $I$ and $U$ and the PV yield $PV_y$ is derived as the integrated MPP over each day

$$PV_y = \int_{day} I_{PH}(t)\, U(t)\, dt. \qquad (4)$$

The linear relation of PV yields and incoming irradiance is used for a simplified linear model for daily PV yield ($PV_y$)

$$PV_y = a(T) \cdot \mathrm{GTI} + b(T). \qquad (5)$$

For our purpose it is sufficient to replace GTI with a normalized GHI ($GHI_{norm}$, also to reduce the seasonal variability) from SARAH-2.1 which is calculated by dividing the GHI with the cosine of the minimum daily zenith angle. Note, that due to the high importance of the cell temperature the fitting parameters (Equation 5) depend on temperature. The parameter $b$ indicates the impact of the inverter, as it needs a certain amount of power to work. The slope $a$ indicates the efficiency, including the conversion of $\mathrm{W/m^2}$ to kilowatt hours per kilowatt-peak ($\mathrm{kWh/kWp}$). Uncertainties due to a varying temperature and the
coefficients $a_i(T)$ and $b_i(T)$ will be estimated by calculating the explicit PV power, including temperature, at three sites and its variability (see Section 3.2).

To determine $a_i(T)$ and $b_i(T)$ explicit PV power calculations are undertaken by using the PV power model part of the "Solar Power modeling including atmospheric Radiative Transfer" (SolPaRT) model at Agoufou (Mali), Banizoumbou (Niger) and Djougou (Benin) at 15-minute resolution (Neher et al., 2019). These calculations require the knowledge of the incoming
radiation on the tilted plane and cell temperature over the diurnal cycle. These parameters can be derived by using the GHI, DIR, the solar zenith angle, the ambient temperature and wind speed. The impact of soiling and shading is excluded here, as it highly depends on local conditions and the cleaning cycle of the modules. For the explicit calculations, the SARAH-2.1 data record of GHI, depending on the solar zenith angle, and the modules orientation (latitude assumed as the tilt and southern orientation) are used to determine the radiation on the tilted plane. Assuming an installation with eleven modules (typical size
of one row in a PV plant, several can be connected in parallel) the inverter is only slightly (96%) over dimensioned, as high irradiance is expected in the considered region. The input data for the model calculations, including sources, are summarized in Table 2.

**Table 2.** Input data for photovoltaic power calculations.

| Name | Value | Resolution | Type | Source |
|---|---|---|---|---|
| GHI | continuous | 1/2 hourly | linear interpolation | SARAH-2.1 |
| Ambient temperature | continuous | 15 min | | AMMA |
| Wind speed | continuous | 15 min | | AMMA |
| Tilting angle | latitude | | | definition |
| Orientation | South | | | definition |
| Cell material | | | silicon | definition |
| No. of modules | 11 | | | definition |
| Inverter | | | SMA 2500U | definition |

## 3.2 Uncertainties of PV yield estimation

The PV power is explicitly calculated (using Equation 1, the temperature information from AMMA and the GHI and DIR from SARAH-2.1) at the three measurement sites (Agoufou, Mali; Banizoumbou, Niger; Djougou, Benin) in a 15 minute resolution. For each day, the PV yield (integral over each day and normalization over the plant peak - given in $\mathrm{kWh/kWp}$) is derived. On a daily basis the GHI itself depends on atmospheric conditions (clouds, aerosols and greenhouse gases) and season (solar zenith angle). PV yields are highly correlated to the daily mean normalized GHI (see Figure 3). However, the

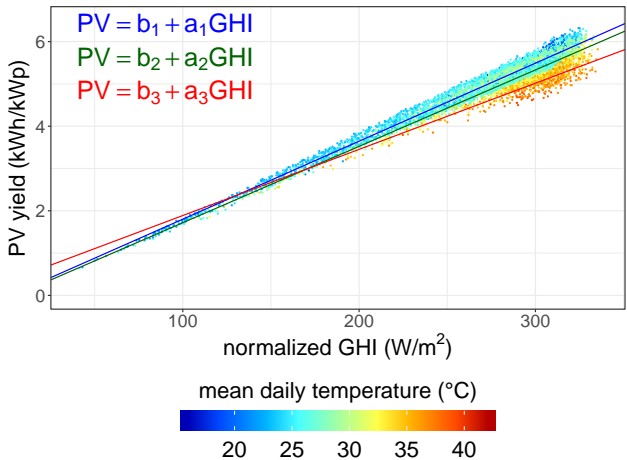

**Figure 3.** PV yield as a function of normalized global horizontal irradiance combining the calculations at all three measurement sites at three temperature levels, $T \leq 30^{\circ}C$ (blue), $30^{\circ}C < T \leq 35^{\circ}C$ (green), $T \geq 35^{\circ}C$ (red). The mean daily temperature is marked as color.

mean daily temperature additionally impacts PV yields. Therefore, three regression lines (see Equation 5) are determined at different temperature levels ($T \leq 30^{\circ}C$ (blue), $30^{\circ}C < T \leq 35^{\circ}C$ (green), $T \geq 35^{\circ}C$ (red)). For a finer separation into more temperature classes no further significant improvement was found. The mean daily temperature are used from ERA5 to define the temperature level of each point during each time step. The explained variance ($R^2$) is highest for the lowest temperature ranges (0.98) and increases (0.78) for the highest (see Table 3). The root mean square error (RMSE) and $R^2$ as well as the single fitting parameters $a_i$ and $b_i$ are summarized in Table 3.

**Table 3.** Statistical and fitting parameters of PV yield correlation.

| Temperature level | $T \leq 30^{\circ}C$ | $30^{\circ}C < T \leq 35^{\circ}C$ | $T \geq 35^{\circ}C$ |
|---|---|---|---|
| RMSE ($\mathrm{kWh/kWp}$) | 0.16 | 0.25 | 0.18 |
| $R^2$ | 0.98 | 0.89 | 0.78 |
| N | 5244 | 1890 | 474 |
| $a_i$ ($\mathrm{hm^2/Wp}$) | 0.018 | 0.018 | 0.017 |
| $b_i$ ($\mathrm{kWh/kWp}$) | -0.04 | -0.09 | 0 |

The slope $a$ decreases at increasing temperatures. For $T \geq 35°C$ the parameter $b$ was set to zero, as for physical reasons it can not be positive. The uncertainty is highest at the highest temperature level (RMSE: $\pm$ 0.67 kWh/kWp) and lowest at the lowest temperature level (RMSE: $\pm$ 0.16 kWh/kWp). The variability of PV yields increases with the normalized GHI, due to two reasons. First, temperature levels can reach higher values at higher normalized GHI, which would induce a higher reduction of PV yields compared to lower temperature levels. Second, the temperature effect on PV yields is relative and can

reach higher effective PV yield reductions at higher normalized GHI.

## 4    Validation of satellite data with ground-based measurements

Previous studies compared the SARAH-2.1 GHI to ground-based measurements from the BSRN (Pfeifroth et al., 2019b), as they provide benchmarks in accuracy ($\pm 2\%$ or 5 W/m$^2$ for GHI). However, there is currently no BSRN station running in West Africa. Therefore, we use ground-based measurements of GHI from the AMMA campaign at three sites for the satellite

data validation (see Table 1). The comparison of SARAH-2.1 GHI to observed GHI is conducted for daily and monthly means (see Figure 4). Statistical parameters, i.e. $R^2$, root mean square error (RMSE), MAE and bias, are used for comparison.

In Banizoumbou (Figure 4 b, MAE of 15.8 W/m$^2$ for daily and 7.6 W/m$^2$ for monthly means), SARAH-2.1 performance is consistent with previous evaluation against BSRN stations (Pfeifroth et al., 2019b). Similarly, the bias with -0.9% to -1.2%, the $R^2$ with around 0.8 and the RMSE with 20.1 W/m$^2$ for daily and 9.5 W/m$^2$ for monthly mean GHI are in the same order

as those found by Pfeifroth et al. (2019b). However, at the two other sites GHI is overestimated (bias up to 12%). At the desert site (Agoufou) the $R^2$ is only 0.5 for monthly mean GHI. Due to the sandy environment, dust deposition on the measurement equipment might cause errors in the observations of GHI (measurement uncertainties are 2% in Banizoumbou and Djougou and 10% in Agoufou). Furthermore, the instrument maintenance of the measurement equipment is not known and can be a source for additional uncertainties. In Djougou (Savanna site) the overestimation is comparably high with a bias of 12% and MAE

over 25 W/m$^2$. Monthly mean GHI generally show higher accuracy, as the variability is reduced due to averaging reasons.

The three sites provide evidence on a higher MAE in West Africa (up to 27.6 W/m$^2$) compared to the validation with mainly European BSRN stations in Pfeifroth et al. (2019b) (MAE: 11.7 W/m$^2$) for daily GHI. One reason for this deviation could be the generally higher GHI in West Africa (Solargis, 2019) compared to Europe (where most of the used BSRN stations are located). Furthermore, higher aerosol loads, which are not explicitly treated in the satellite retrieval, in West Africa compared

to the rest of the world could also cause the deviation.

To study whether deviations from the climatological AOD used in SARAH-2.1 (see Figure 1 d) might explain the deviation we investigate the impact of the difference between the measured AOD and the climatological AOD for the satellite data retrieval ($\Delta$AOD). A higher overestimation of GHI was found at higher $\Delta$AOD at all sites (up to 100 W/m$^2$ for 1 < $\Delta$AOD < 2; see Figure 4, colors in scatter plots and right column). Thereby, the correlation between $\Delta$AOD and $\Delta$GHI (being the

difference between observed and satellite GHI), shows an overall overestimation at higher $\Delta$AOD in Agoufou and Djougou, while in Banizoumbou in some situations a underestimation is visible. The abscence of a systematic effect raises our confidence in using the satellite date to provide an overview on PV potential in West Africa.

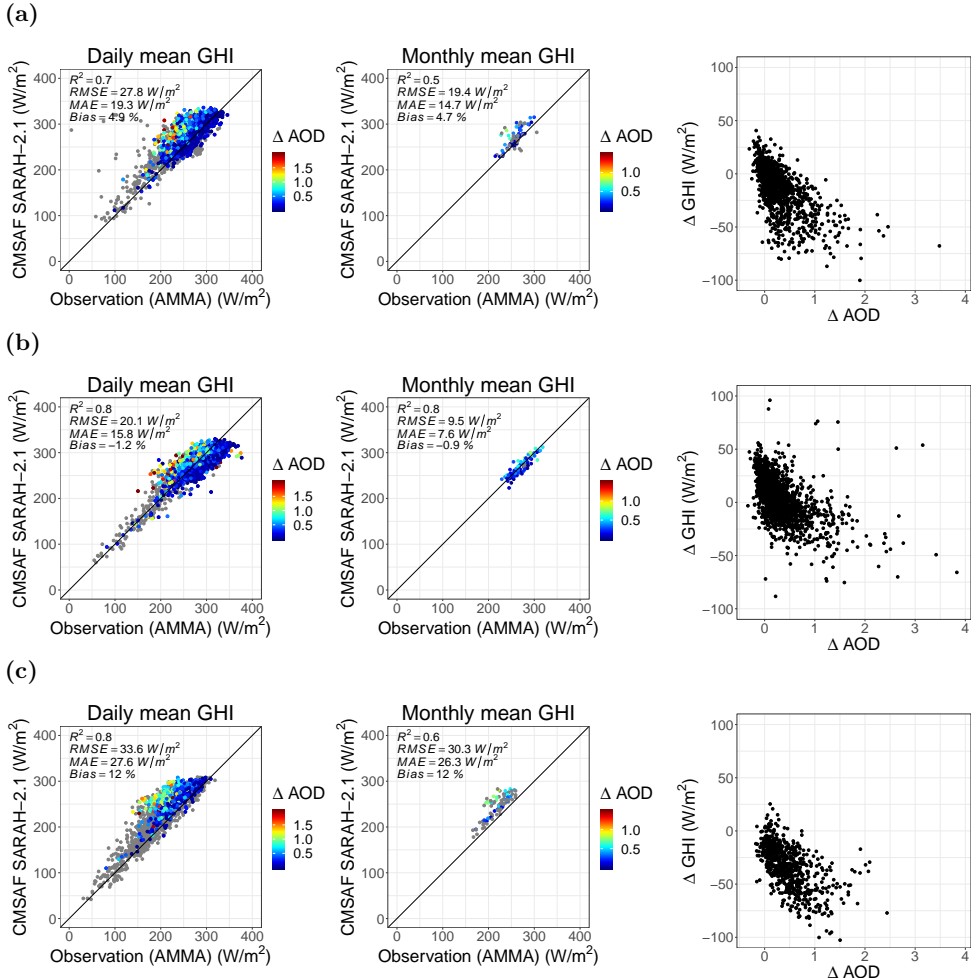

**Figure 4.** Comparison of simulated and observed GHI as daily (left) and monthly (right) averages at three sites over the given timely horizon, a) Agoufou (2005-2008), b) Banizoumbou (2005-2012) and c) Djougou (2002-2009). The difference between the measured AOD and the climatological AOD for the satellite data retrieval (ΔAOD) is indicated as color. If no measured AOD is available, the points are grey. The third column shows the correlation between ΔAOD and the difference between GHI (ΔGHI: observed GHI - satellite GHI).

As the climatological AOD, used in the SARAH-2.1 retrieval, shows values between 0.15 and 0.3 (see Figure 1 d), high ΔAOD (e.g. above 0.5) are connected to high aerosol loads (e.g. dust outbreaks, biomass burning (Marticorena et al., 2011)). Thus, the missing explicit treatment of AOD in the satellite retrieval could be a reason for the low accuracy here. Especially during events with high aerosol loads an explicit treatment in the SARAH-2.1 data retrieval could improve the accuracy of GHI. By using only values with ΔAOD < 0.5 the RMSE is reduced by around 1% to 30% (Agoufou: 29% for daily and 25% for monthly GHI, Banizoumbou: 6% for daily and 1% for monthly GHI, Djougou: 13% for daily and 30% for monthly GHI).

Ineichen (2010) compared ground-based measured GHI to different satellite products in Africa for the single year 2006, including several AMMA sites in West Africa, and found standard deviations between 12% and 37% as well as a bias between -1% and 11%. These values lie in a similar range to our calculations. However, especially during the West African Summer Monsoon low-level clouds are likely not realistically represented in satellite products and climate models in southern West Africa (Hannak et al., 2017; Linden et al., 2015). Hannak et al. (2017) found an overestimation in GHI of up to 50 W/m$^2$ in the rainy season (July to September) for SARAH-1 data in comparison to measurements in southern West Africa between 1983 and 2008. In Kothe et al. (2017) monthly sums of sunshine duration from SARAH-2 (1983-2015) were compared to Global Climate Data (CLIMAT) (Deutscher Wetterdienst, 2019) in Europe and Africa. At several stations in West Africa they found an overestimation of more than 50 h of satellite based monthly sunshine duration compared to CLIMAT. The majority of the CLIMAT stations are located on the southern edge of the Sahel region or south of it. Thus, the findings are especially relevant for southern West Africa. Pfeifroth et al. (2019b) analyzed the accuracy of the SARAH-2 data record for Europe and found a slight decadal but stable trend of -0.8 W/m$^2$.

Given these results, we conclude that the SARAH-2.1 data record can be used to get an overview on the temporal and spatial irradiance variability as well as on trends to estimate the PV potential in West Africa. However, especially in southern West Africa the systematical overestimation of solar irradiance in the SARAH-2.1 data set (Kniffka et al., 2019; Hannak et al., 2017) need to be considered in the conclusions of the variability and trend analysis. As a consequence of the positive offset in southern West Africa, the actual north-south gradient in the satellite data set is underestimated. In particular, for the trend analysis the systematic offset would not have an impact. Overall, an expansion of measurements over longer time periods (the measured data is available for less than 20% of the time period at only three sites) could increase the significance of our validation.

## 5 Changes of solar irradiance

In this section the temporal and spatial variability of GHI and DIR is analyzed for West Africa (latitude: 3°N to 20°N and longitude: 20°W to 16°E) over a 35-year time period (1983-2017). Therefore, the temporal mean and its interquartile range (IQR, identifying the range for 50% of the data with the 25% and 75% quantile as borders) are derived. The analysis is conducted based on the daily values. For GHI the analysis is expanded for the dry and the wet seasons separately. Furthermore, a trend analysis is undertaken for GHI by assuming a simple linear trend based on annual values. The significance of the trend is checked by calculating the 95% confidence interval. The trends are significantly positive (negative) if the upper and lower value of the 95% confidence interval are positive (negative). At four locations, distributed over different latitudes, a time series analysis is additionally undertaken. Monthly means and monthly anomalies are derived for all seasons separately.

### 5.1 Spatial analysis

The spatial distribution of annual mean GHI and DIR are shown in Figure 5. For each grid point also the IQR of all daily mean values is provided for GHI and DIR. The irradiance is high in the Sahel zone and the Sahara (with GHI > 250 W/m$^2$ and DIR around 200 W/m$^2$ north of around 13°N, see Figure 5 (a) and (b)). Towards the southern coast the irradiance decreases, as the

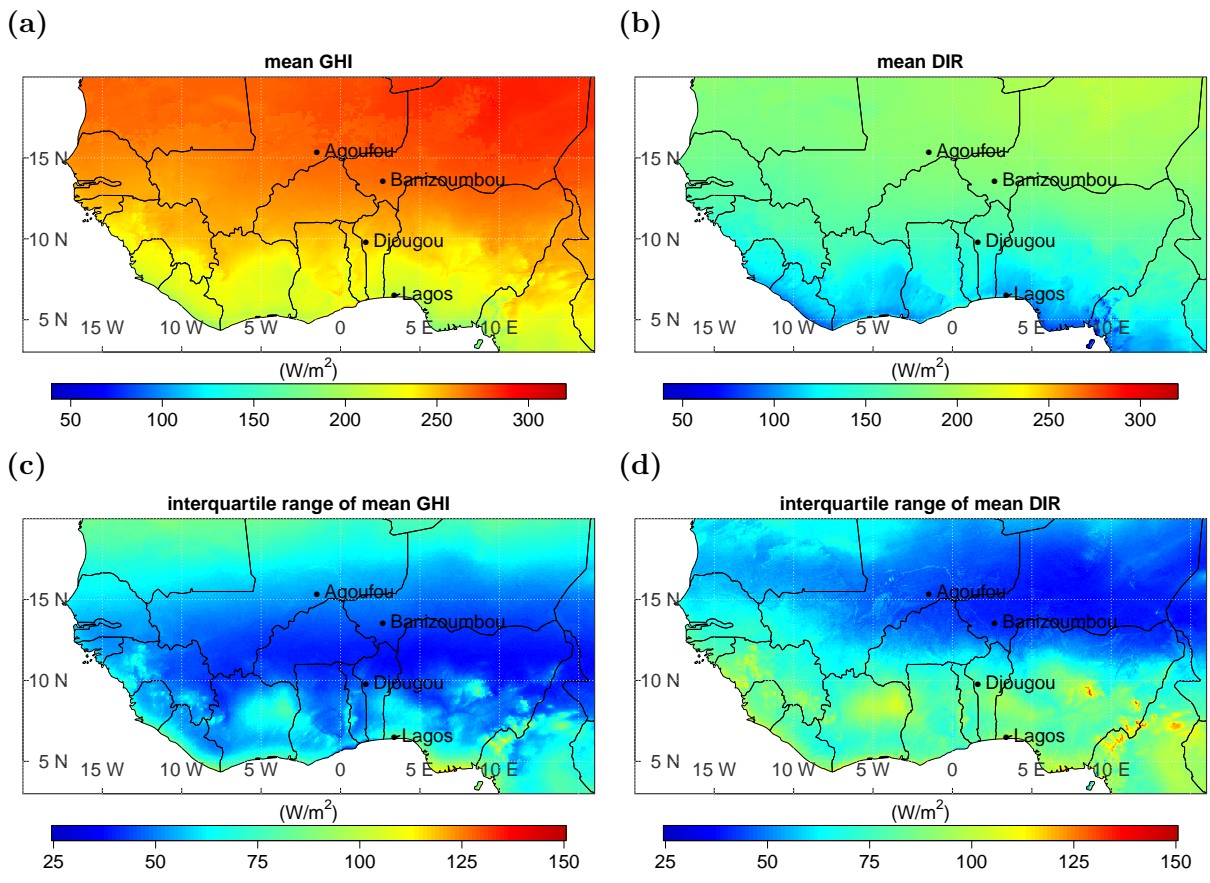

**Figure 5.** Mean (1983 to 2017) global (a) and direct horizontal irradiance (b), with their temporal interquartile range (c) and (d).

cloud cover increases (see Figure 1 b). The impact of clouds seem to be especially high in southern West Africa, south from the Sahel Zone. However, the satellite retrieved GHI might even be overestimated in this region (see Section 4). The temporal variability is higher in southern West Africa (locally up to 150 W/m$^2$) than in the Sahara and the Sahel zone (higher IQR in the South compared to the North), especially for the DIR in coastal or mountainous regions and typical for variable cloud conditions. The impact of clouds on DIR is higher than on GHI, as forward scattered light on cloud droplets is still included in the GHI but not in the DIR. The high amount of water vapor in coastal regions could favor the formation of clouds and could therefore be a reason for the higher variability of DIR. Furthermore, the wet season is actually longer in southern West Africa than in the northern parts (CLISS, 2016), which leads to longer periods with high cloud cover and could be further favored by orographic cloud development (see Figure 1 b). However, the same analysis with a more confined definition of seasons (dry: November - March, wet: May - August) leads to similar results.

When looking at the dry and wet season separately, the spatial GHI distribution reveals a complementary structure (see Figure 6, including the difference to the temporal IQR). For GHI a sharp line at around 13°N to 14°N divides the northern

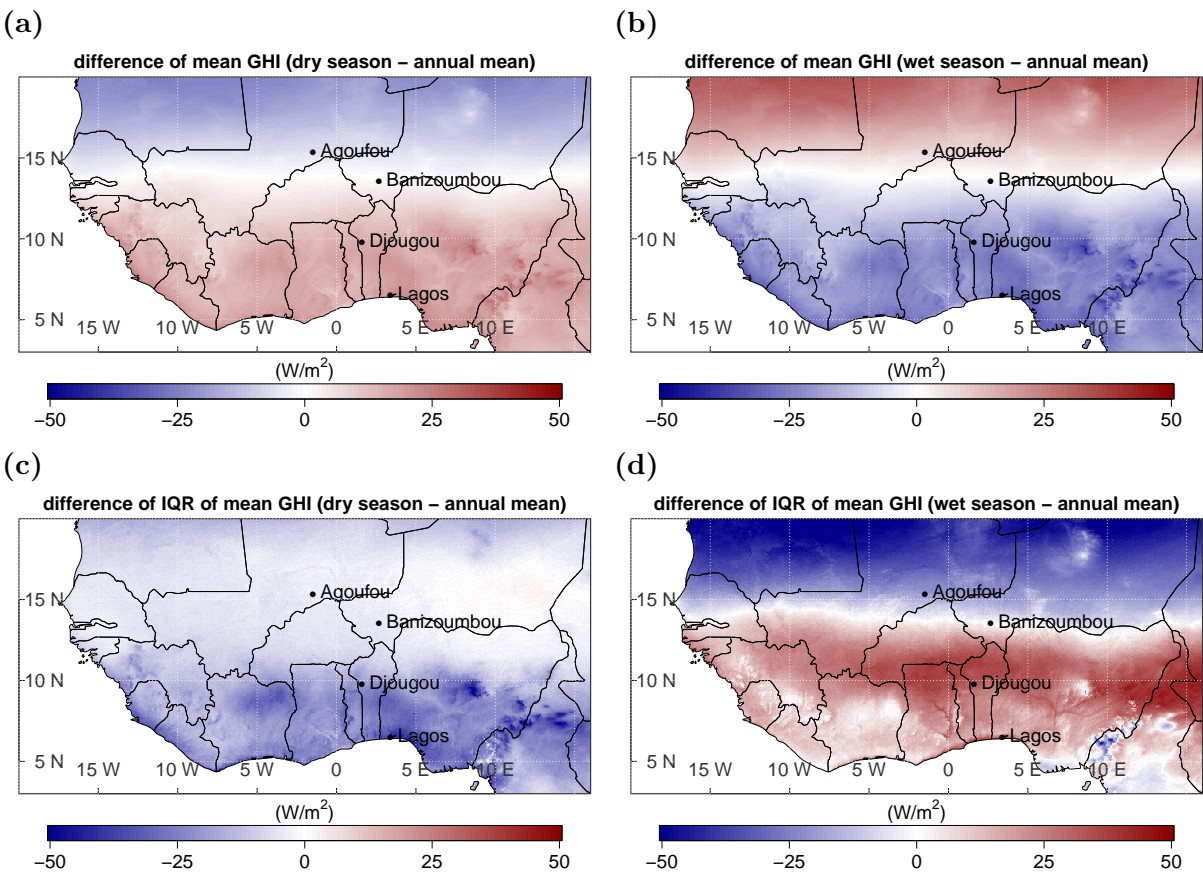

**Figure 6.** Difference of mean global horizontal irradiance during the dry (a) and wet season (b), each to the annual mean, with the difference of its interquartile range (c) and (d) to the interquartile range of the annual mean.

region of the Sahel zone and the Sahara from southern West Africa. North of this line, the GHI is lower than the annual mean (up to -26 $\text{W/m}^2$) during the dry season and higher (up to +36 $\text{W/m}^2$) during the wet season. The northern region experiences

low cloudiness throughout the year (the mean effective cloud albedo is lower than 0.1 in the major part of this region, see Figure 1 b). Therefore, the irradiance mainly depends on the solar zenith angle, which is lower during the wet season than during the dry season. Lower solar zenith angles result in higher surface irradiance under clear sky conditions. In southern West Africa (south of 13°N) GHI is higher (up to +33 $\text{W/m}^2$) during the dry season and lower (up to -46 $\text{W/m}^2$) during the wet season, compared to the annual mean. Cloudiness is comparably high in this region (with a mean effective cloud albedo

up to 0.3, see Figure 1 b). Therefore, clouds are the major modulator of solar irradiance here. As clouds predominantly occur during the wet season, the GHI is lower during this season.

The difference in the temporal variability is given as the difference of IQR (season - annual mean). During the dry season, the temporal variability of GHI shows an overall reduction over land compared to the annual mean. However, in southern West

Africa the reduction goes up to more than -50 W/m² while in the northern part the reduction is hardly visible. During the
wet season, the temporal variability of GHI shows the same sharp boundary at around 13°N to 14°N as the GHI itself but
vice versa. The temporal variability is lower (reaching more than -50 W/m² difference) in northern West Africa and higher
(reaching more than +50 W/m² difference) in southern West Africa compared to the annual mean. This variability is mainly
driven by the WAM, occurring during the wet season (Sultan et al., 2003).

The regional mean and its IQR (concerning the spatial variability) for GHI and DIR are summarized in Table 4. The regional

**Table 4.** Regional mean and regional interquartile range (IQR) of the temporal mean GHI and DIR between 1983 and 2017 for the annual mean, the dry and the wet season.

|  | Annual mean | | Dry season | | Wet season | |
|---|---|---|---|---|---|---|
|  | mean | IQR | mean | IQR | mean | IQR |
| GHI (W/m²) | 250 | 37 | 254 | 20 | 246 | 67 |
| DIR (W/m²) | 159 | 45 | 169 | 34 | 145 | 68 |

variability of solar irradiance is higher during the wet season compared to the dry season, as clouds, predominantly occurring
during the wet season, have a large impact on solar radiation. During the wet season, the regional variability lies in a similar
range for the GHI as for the DIR, with an IQR of 67 W/m² for GHI and 68 W/m² for DIR. As the mean DIR is smaller
than the mean GHI the percentage variability is higher for DIR. This is a clear sign for variable cloudiness leading to a higher
variability of diffuse irradiance.

## 5.2 Temporal analysis

The results show strong gradients between North and South as well as the wet and the dry season. To detect anomalies and
changes in variability within the north-south axis, four locations are chosen for a time series analysis of the SARAH-2.1 data
record (the three measuring sites from Section 4 and one coastal location (Lagos, Nigeria - 6.5°N; 3.4°E), see Figure 7). The
respective data record (see Section 2.1) is used between 1983 and 2017 in a daily resolution.

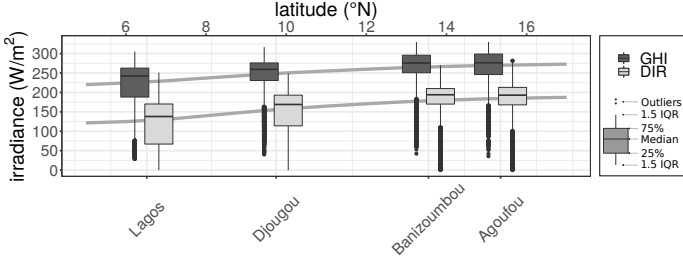

**Figure 7.** Median daily global (dark gray) and direct horizontal irradiance (light gray) from the SARAH-2.1 data set at Agoufou, Mali;
Banizoumbou, Niger; Djougou, Benin and Lagos, Nigeria as a function of latitude. The variability is illustrated by box plots showing the
interquartile range and whiskers. The gray line connects the mean GHI (top) and DIR (bottom) over each latitude of the study region.

The median GHI and DIR decline with decreasing latitude (see also Figure 7) while their variability increases with decreasing latitude (as the IQR increases). The higher frequency of clouds in southern West Africa likely drives this variability. At the desert and Sahel locations (Agoufou and Banizoumbou) the IQR of the GHI is larger than the IQR of DIR. Thus, the variability is higher for GHI than for DIR, while it is the opposite at the southern locations (Djougou and Lagos).

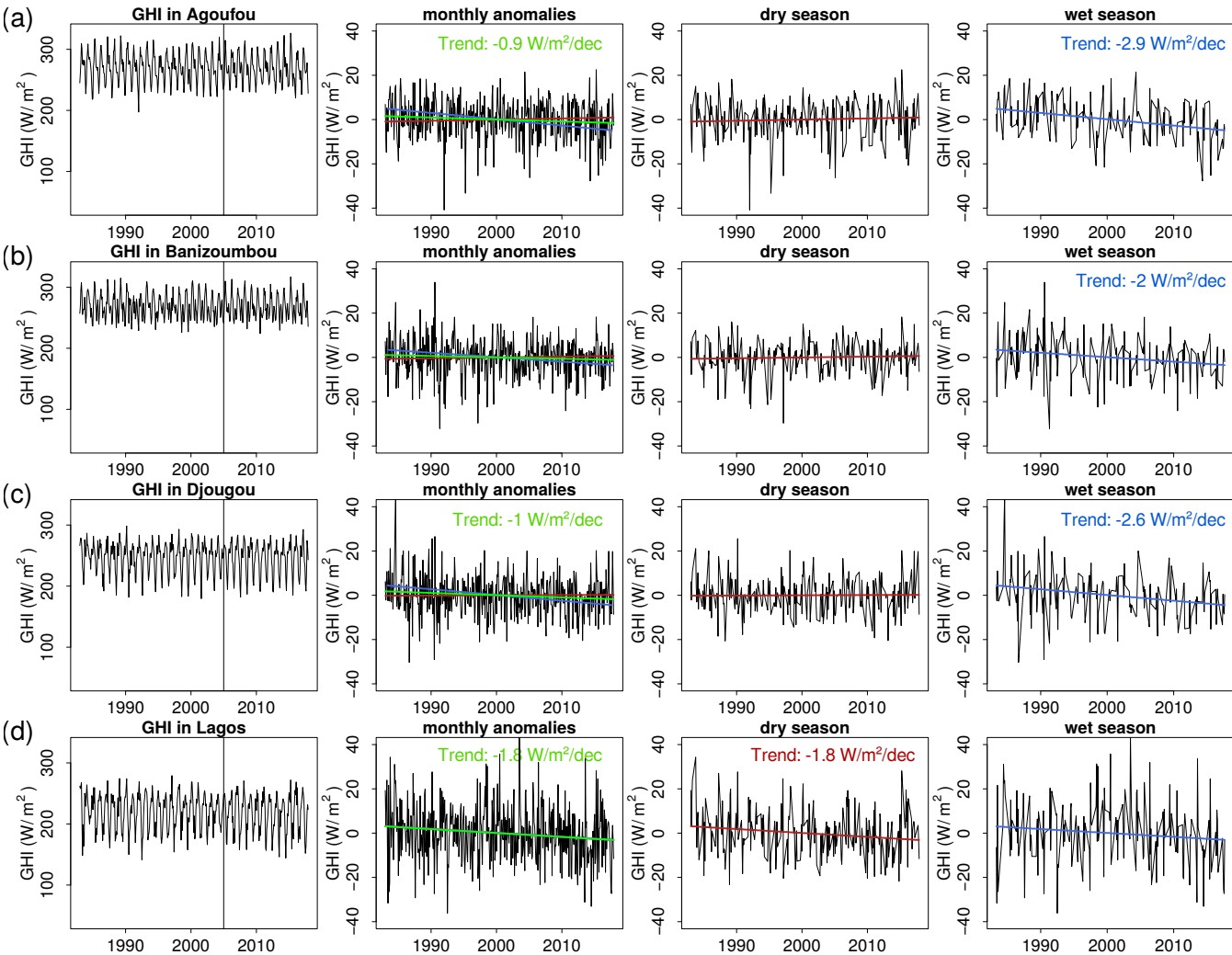

**Figure 8.** Satellite based time series of monthly mean global horizontal irradiance with their monthly anomalies and trends for the annual mean (green), as well as during the dry (red) and wet (blue) season separately in Agoufou (a), Banizoumbou (b), Djougou (c) and Lagos (d). The linear trend of the anomalies is shown monthly, as well as for the dry and wet season separately. Trends are quantified in the single plot windows if they are significant (p-value < 0.05). The black vertical line indicates the time of the change from the Meteosat first to second generation satellites.

For a more detailed look, time series of monthly mean GHI and DIR and their anomalies are pictured for the four locations in

Figure 8. At the southernmost location (Lagos) the trends in anomalies are similar for the wet and the dry season (negative trend of -1.8 W/m$^2$/decade). At all the other locations the dry season anomalies are rather constant (showing no significance) while the wet season anomalies shows decreasing significant trends (ranging from -2 W/m$^2$/decade to -2.9 W/m$^2$/decade) which provides a significant trend over the full year for Agoufou, Djougou and Lagos. At all locations, no significant breakpoint can be seen for the change from Meteosat first to second generation satellites in 2005.

As long-term changes in climate conditions (e.g. temperature, precipitation) have been found over the entire region (Barry et al., 2018), the trends of global irradiance over the last 35 years are analyzed (see Figure 9) as they are of high importance for a future PV system. Especially during the wet season mean temperature increased along the coast between 1983 and 2010 (Yaro and Hesselberg, 2016). The general positive trend of temperature over the region can be found in the ERA5 data as well, with up to 0.22°/decade.

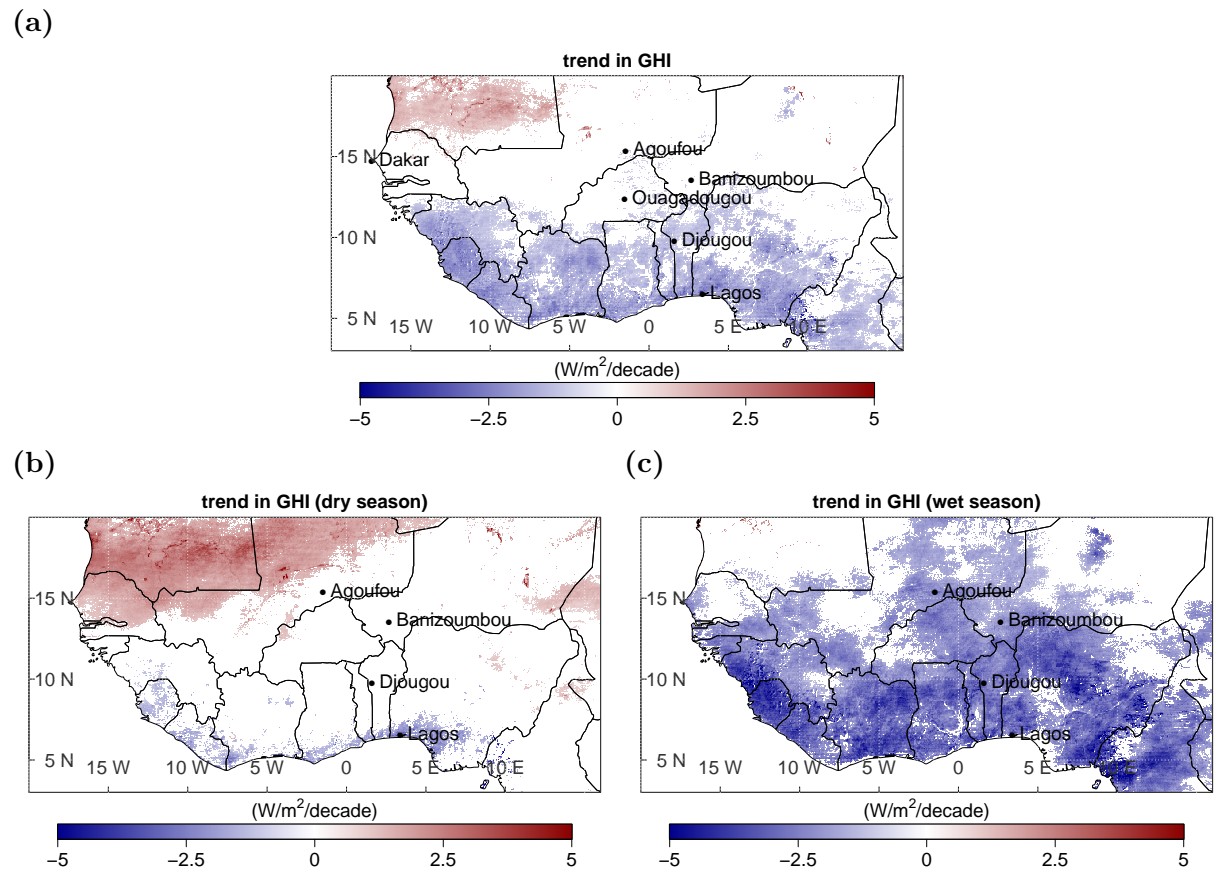

**Figure 9.** Linear trend for global irradiance of the annual mean (a), as well as the dry (b) and the wet season (c), each for all significant cases (based on the 95% confidence interval). Ouagadougou, Burkina Faso and Dakar, Senegal are additionally visualized here, as values at these locations are compared within this section.

Trends of GHI during the time period 1983 - 2017 are positive in the West African Sahara and negative south of the Sahel zone. By looking at the dry and wet season separately, the major part of the negative trend can be attributed to the wet season. The positive trend occurs mainly during the dry season. Overall, the decadal trends are small (in the range of 1% - 2% per decade) compared to the absolute surface irradiance as well as the IQR. However, the absolute values of the trend reach around $\pm 5 \ (\mathrm{W/m^2})/\mathrm{decade}$ and are significant (based on the 95% confidence interval). Compared to the uncertainties of the satellite data (MAE up to 27.6 $\mathrm{W/m^2}$, see Section 4), the trends might seem negligible. However, the reported uncertainties are not bias corrected and represent, in particular in the case of Djougou, the systematic overestimation of the GHI by the satellite estimate. The estimation of the temporal trend is unaffected by any systematic over- or underestimation and, hence, still can be derived with certain confidence.

The negative trend south from the Sahel region indicates an increasing cloud cover or a higher amount of water vapor in the air. Especially low level clouds are frequent during the wet season in southern West Africa (Linden et al., 2015). These clouds were analyzed during the Dynamics–aerosol–chemistry–cloud interactions in West Africa (DACCIWA) campaign in 2016 (Knippertz et al., 2015). They form at night and are present during the day with a peak in cloudiness in the mornings. Local aerosols can increase the cloud droplet number concentration by 13% - 22% (Taylor et al., 2019), brightening the clouds and reducing the GHI. The southern regions of West Africa were affected by agriculture expansion and urbanization in the last decades (CLISS, 2016). This leads to a higher portion of local aerosols in the atmosphere which can serve as cloud condensation nuclei that foster cloud formation and cloud optical properties. Furthermore, a positive trend in water vapor was found on the coast of tropical oceans in West Africa from satellite data (Mears et al., 2018), which would reduce the GHI at the surface.

The positive trend in irradiance in the Sahara might be driven by the reduction of dust movement, which was found in several data sets since the 1980s (Cowie et al., 2013). Furthermore, a reduction of cloudiness could be a reason for the increasing irradiance.

The detected trends are in the range of global dimming and brightening tendencies (-9 to +4 $(\mathrm{W/m^2})/\mathrm{decade}$), which originate from atmospheric changes (caused by e.g. anthropogenic pollution and visible due to aerosol variation and aerosol-cloud interactions) (Wild, 2012). The mentioned trends in cloud occurrence could be driven by a change in the WAM, the Hadley circulation and water vapor as well as the shift of the ITCZ (Byrne et al., 2018; Roehrig et al., 2013). Furthermore, also aerosols can play a decisive role. As mentioned before, aerosols are highly variable in the West African region and can reach extreme values (AOD up to 4). In the satellite data retrieval aerosols are not treated explicitly. Therefore, their variability can cause high uncertainties for the trend analysis, especially during the dry season, when clouds are mostly absent and aerosols are the major modulator of GHI. Furthermore, in West Africa different aerosol types are present in the atmosphere (e.g. dust, marine, anthropogenic and biomass burning aerosols), which differ in their atmospheric impact on GHI (absorption and scattering, spectral dependence). However, Neher et al. (2019) found, that the impact of AOD is around four times higher compared to the impact of the aerosol composition on PV power in West Africa. Yoon et al. (2012) found a negative trend in AOD for Dakar, Senegal (1996 - 2009) and Ouagadougou, Burkina Faso (1995 - 2007) and a positive trend in Banizoumbou, Niger (1995 - 2009). The detected trends in GHI from SARAH-2.1 data (1983 - 2017) at these locations are negative in

Banizoumbou and positive in Dakar. Thus, changes in aerosols could be a major driver for the trends in GHI. However, in Ouagadougou, the trend in GHI is negative. Thus, other meteorological changes, e.g. clouds, might be larger than the trend in AOD. In general, trend analysis is a complex topic. However, a clear regional distribution might enable us to better identify the causes for the trends when looking at PV power.

## 6 Implications for photovoltaic yields

Photovoltaic yields are calculated for each day over the whole region by using the linear model (Equation 5) with the parameters derived in Section 3.2 for each temperature range (see Figure 10 for mean PV yields and temporal IQR). The temperature level is taken from ERA5 as daily means. As we used a linear approach, the uncertainty of satellite data would propagate linearly for PV yield estimates.

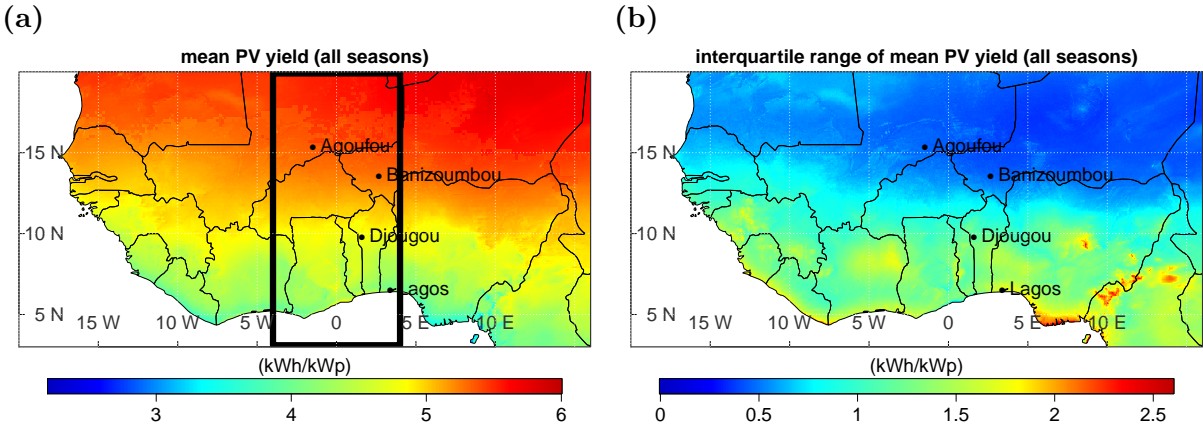

**Figure 10.** Annual mean (1983 to 2017) PV yield (a) and its interquartile range (b) over the full region. The black box in (a) marks the longitude range for Figure 11.

As a result of using a linear regression to derive PV yields, the temporal variability of PV yields (mean: $4.9\,\mathrm{kWh/kWp}$, IQR: 20%) is lower compared to the temporal variability of GHI (mean: $250\,\mathrm{W/m^2}$, IQR: 24%). However, the regional variability is 3 percentage points higher for PV yields (IQR: 18%) than for GHI (IQR: 15%). Here we go a step further and analyze the regional variability over each latitude (in the longitude range between 4°W and 4°E to exclude ocean regions), annually as well as for the dry and wet seasons separately. Figure 11 shows the variability of the temporal mean PV yield for each latitude separately.

The explicitly calculated PV yields at Banizoumbou and Djougou lie in the variability range of the corresponding latitude demonstrating the appropriateness of the simplified model for PV calculations. However, the most northern site, Agoufou is lower than the daily modeled data at 15°N. A possible reason might be due to the high temperatures encountered here. The uncertainties of the linear model are highest for high temperatures (RMSE: $0.67\,\mathrm{kWh/kWp}$, see Table 3). In the northern

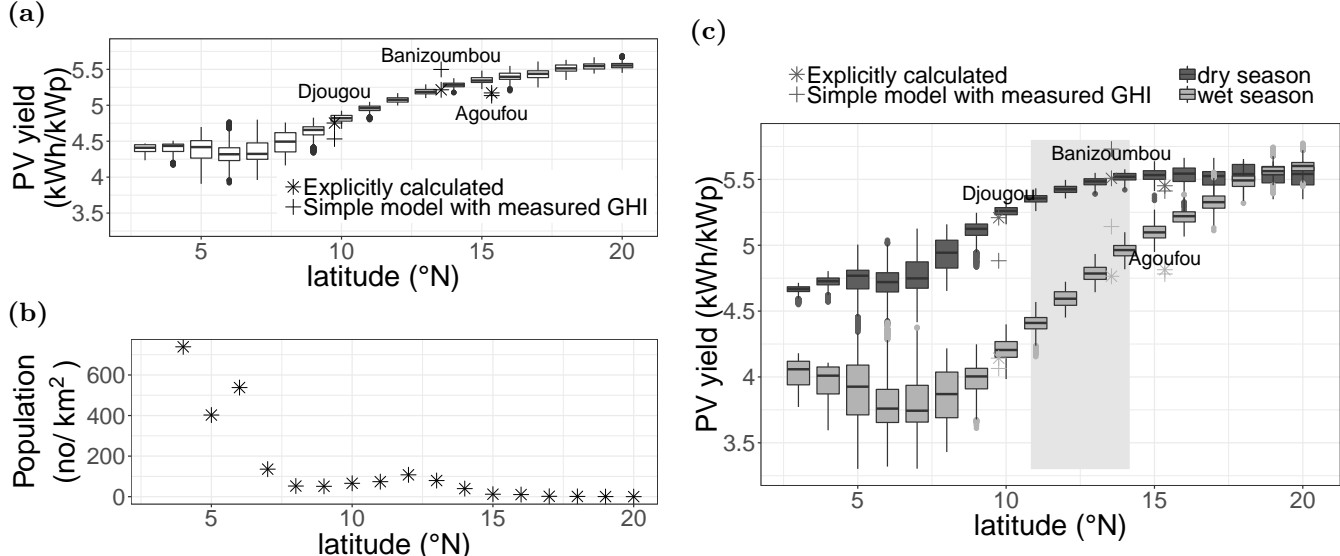

**Figure 11.** Mean (temporal) PV yield at each latitude, for the total year (a), population density for each latitude (b, (NASA, 2020)), as well as mean PV yield at each latitude for the dry: October-April (light grey) and wet season: May-September (dark gray) (c), in the longitude range between 4°W and 4°E. The single points mark the temporal mean PV yield calculated with the explicit model and measured ambient temperature (star) as well as the PV yield calculated with the simple model and measured GHI (cross) at the three sites, Agoufou (2005-2008), Banizoumbou (2005-2012) and Djougou (2002-2009). The gray background box in (c) marks the latitude range, where the definition of seasons is most accurate.

part of West Africa the monthly mean temperature can reach more then 40°C (Berrisford et al., 2011). Thus, the PV yields at high latitudes could actually be lower. Furthermore, the PV yield at each site is calculated with ground based measured temperatures, while the model uses daily temperatures from ERA5. At Agoufou the averaged daily mean of the ground based

380  temperature over the total time span is around 3.5°C higher than the mean ERA5 temperature.

In general, the overestimation of satellite data in Agoufou and Djougou as well as the slight underestimation in Banizoumbou (see Section 4) can be seen in the PV yields calculated with the simple model and using the measured GHI as an input (crosses in Figure 11 a). In Agoufou, the PV yields, calculated with the linear model, are similar to the explicitly calculated PV yields. In Banizoumbou the results are higher and in Djougou they are lower compared to the PV yields calculated with satellite data.

385  Especially in Djougou, the irradiance decreases over the 35 years of satellite data availability. This leads to lower values in the 2000's values compared to the mean.

By looking at the full latitude range, PV yields are smaller at low latitudes (around 4.5 kWh/kWp) with a higher regional variability and more outliers. At high latitudes, the PV yields reach around 5.5 kWh/kWp, which is around 22% higher than at low latitudes. Furthermore, an overestimation of solar irradiance was found in southern West Africa (see Section 4). Thus,

the spacial gradient between North and South could actually be higher than suggested by the satellite estimation. Population density shows the opposite latitudinal gradient compared to PV potential, with a higher density at low and a lower density at high latitudes (see Figure 11 b).

During the dry season PV yields are similarly spread over the different latitudes than the annual PV yields. However, the yields are slightly higher (by around 0.25 kWh/kWp) at low latitudes (between 3°N and 10°N). During the wet season a band of lower PV yields (less than 4 kWh/kWp) is visible between 3°N and 9°N. This is the region, where low level clouds occur frequently (Linden et al., 2015).

## 7 Challenges for the West African power sector

Currently, there exists a deficit between power demand and supply in West African countries (Adeoye and Spataru, 2018). Furthermore, up to 2030 the power demand may increase to the fivefold of the 2013 demand (IRENA, 2015). Thus, new large scale power plants need to be developed and the infrastructure needs to be built up. The West African Power Pool (WAPP) was founded in 1999 to coordinate these developments. The business plan of the WAPP plans the connection of 14 countries with high voltage transmission until 2025 (WAPP, 2015). Especially photovoltaic (PV) power is expanding, with a technical potential of around 100 PWh/year (Hermann et al., 2014) and has high expectations to meet a large share of future power supply (IRENA, 2015). Therefore, the long-term changes in PV power potential are relevant and addressed in this study.

Solar irradiance is the key driver of photovoltaic power potential. The dimension and built up of new power plants requires a specific site analysis of solar irradiance to estimate expected economic benefits. Thereby, long-term changes as well as the day to day variability need to be taken into account to dimension the plant, the necessary storage capacities and to design the grid. In this study, 35 years of satellite based irradiance data (the SARAH-2.1 data record) is locally validated and used to get a spatially complete distribution of photovoltaic potential over West Africa (3°N to 20°N and 20°W to 16°E).

In summary and as expected, there is a strong contrast in photovoltaic potential during the dry and wet season, controlled by the West African Monsoon (WAM) and the accompanied seasonal movement of the Inter Tropical Convergence Zone. The dry season provides higher photovoltaic yields than the wet season, especially in southern West Africa (dry: around 4.75 kWh/kWp; wet: down to 3.75 kWh/kWp). Furthermore, a strong contrast can be seen between the higher potential in the northern (up to 5.5 kWh/kWp) and the lower potential in the southern parts of West Africa (around 4.5 kWh/kWp). The temporal variability is higher in the south and lower in the north of West Africa as a result of the WAM. Generally, the variability is more pronounced for photovoltaic potential than for global horizontal irradiance, as additional impacts of the inverter reduce the yields of a PV power plant by a certain threshold.

In the Sahara and Sahel zone, daily average global horizontal irradiance reaches up to 300 W/m$^2$ and shows a positive trend of up to around +5 (W/m$^2$)/decade. The opposite trend (with up to around -5 (W/m$^2$)/decade) and lower irradiance is found in southern West Africa, with daily average global horizontal irradiance below 250 W/m$^2$. The trends lie in the range of global dimming and brightening tendencies. Furthermore, the temporal variability is higher in southern West Africa (reaching an interquartile range (IQR) of up to 150 W/m$^2$ in mountainous areas) than in the Sahara and Sahel zone (were the IQR stays

below 100 $\text{W/m}^2$). For direct horizontal irradiance the difference between northern and southern West Africa is similar to the difference in global horizontal irradiance. However, especially in the mountainous region in Nigeria, the temporal variability
is more distinct for direct than for global horizontal irradiance.

     Regarding seasons, there is a sharp difference between the wet and the dry season. During the dry season, average solar irradiance and its IQR are rather constant (global irradiance around 254 $\text{W/m}^2$ and IQR around 20 $\text{W/m}^2$), while during the wet season average solar irradiance varies over the region (with higher values in the north than in the south) and an IQR of around 67 $\text{W/m}^2$. Compared to the annual values, the dry season provides higher global horizontal irradiance in the south and
lower in the north, while the opposite was found during the wet season. Thereby a dividing line at about 13°N can be drawn to separate the south from the north concerning daily variability. This seasonal shift is particularly visible at low latitudes (higher urban density than at high latitudes). This seasonality is dominated by the moist monsoon winds, going along with high cloudiness and coming from the south-west during the wet season and the dry Harmattan winds from the north-east during the dry season. To overcome such seasonal differences in power generation, a smart combination with other power sources (e.g.
hydro power and wind) is necessary, as long-term storage is expensive.

     By looking at the mentioned characteristics, the development of PV power plants is more likely in northern West Africa, as higher yields can be reached. However, more power is consumed in the southern parts of West Africa, close to the coast, where the population is higher. A power generation in the north would therefore reiterate the necessary grid development on a north-south axis to transport the power from the insolation rich Sahara to the urban regions in the south. Larger investigations
on PV systems in the south instead would evoke the development of large storage capacities to compensate fluctuations in PV power generation due to the higher variability of solar irradiances in the South compared to the Sahel zone and Sahara. However, the combination with other renewable power sources (e.g. wind and hydro power) could reduce the needed storage capacities (Sterl et al., 2020). The difference in north-south potential increased over the last 35 years. If this trend is ongoing in the future, the potential PV power in southern and northern West Africa might differ even more. This should be considered
in future grid planing.

     Besides the constant seasonal and intraday variability, extreme events can affect power generation drastically. Major dust outbreaks occur frequently during the dry season in the Sahara and Sahel Zone and can bring reductions in power generation of up to 79% over several days (Neher et al., 2019). For such events storage capacities for several days might be needed e.g. in solely solar based micro grids.

This analysis provides an overview on the photovoltaic potential in West Africa. However, the explicit modeling of a photovoltaic power module at a higher temporal resolution could better resolve the impact of temperature and the inverter for each grid point. Furthermore, to dimension the grid and needed storage capacities explicitly, a demand-supply power model including all available power sources is necessary. This should be subject of further research.

*Author contributions.* IN performed the data analysis and was responsible for the development of the paper. JT and UP provided the CMSAF data and gave advise to the manuscript during the writing process. SC and SM provided the overall scientific guidance, discussed results and gave advice during the writing process.

*Competing interests.* The authors declare that they have no conflict of interest.

*Acknowledgements.* The first author, Ina Neher, is thankful for a PhD fellowship from the Heinrich Böll Foundation. Furthermore, the authors would like to thank numerous data providers: Meteorological data was used from the AMMA Database. Based on an French initiative, AMMA was built by an international scientific group and is currently funded by a large number of agencies, especially from France, UK, US and Africa. It has been the beneficiary of a major financial contribution from the European Community's Sixth Framework Research Program. Detailed information on scientific coordination and funding is available on the AMMA International web site http://www.amma-international.org. We thank Philippe Goloub and Didier Tanre for their effort in establishing and maintaining AERONET sites in Agougou, Banizoumbou and Djougou and provide aerosol data. The CM SAF SARAH-2.1 data record was accessed via www.cmsaf.eu.

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
