# Peer review of "Photovoltaic power potential in West Africa using long-term satellite data"

_Atmospheric Chemistry and Physics, 2020_

## Referee Comment (RC1) · Anonymous Referee #2 · 25 May 2020

General comments

In this paper, the authors use irradiance from satellite data to calculate solar PV yields and variabilities in West Africa and their dependence on latitude, using an empirical linear model and validating their results by comparison to ground-based irradiation data.

Overall, this is an interesting paper that can certainly provide good bases for discussion of the future of solar PV in West Africa, which is likely to be bright (apologies for the lame wordplay) as various recent literature has shown.

The manuscript is generally well-written, although at times it is unclear whether the

authors are talking about observed or satellite data, and the order of the section is not always logical, at least in my eyes. However, if the authors can address this and the specific comments below, I would consider to recommend this article for publication.

Specific comments

Title: I wonder if the long-term variability is the most important output from this paper. Isn't it rather the validated use of satellite data over the region, and the yield-latitude plots in Fig. 10? The authors may, if they agree, reconsider the suitability of the title for the paper.

Line 1: "Long-term changes" -> do the authors mean historical, or future, or both?

Line 2: "Here we use satellite irradiance" -> and temperature from reanalysis, right?

Line 22: "located close to the equator, (...)" -> yes, but in reality, it's the locations furthest from the equator that have the highest PV potential in West Africa, as your research shows.

Line 24: "PV power system" -> this wording occurs at several instances in the paper. What exactly do the authors mean with it? Is it a power system where a certain share of power generation is from solar PV? Or solely based on PV without any other power generation sources? Is there a quantitative definition for it?

Line 35: "no assessment over total West Africa (...)" -> what is meant with "assessment"? Do the authors mean a validation of satellite data? Since this is one of the core pieces of this study, I would recommend the authors to be a lot clearer about the added value of their research here compared to the "no assessment" state-of-play.

Line 42-44: "However, they need (...) certain assumptions." This sentence confuses me – how does it relate to the problem the authors are trying to solve? I thought the focus was long-term changes, but here it sounds as if hourly resolution is the most important problem to be solved by such research.

Line 45: The authors do not really explain here why analysing the long-term changes in West Africa is so important. Is there literature explaining why this is crucial, in particular for solar PV, either for West Africa or for other regions worldwide? Especially as compared to the variability on diurnal and seasonal timescales?

Line 59-60: I have some trouble with the definition of dry and wet season that the authors employ here – the definition seems rather generic for a region spanning a large latitude range. For example, the rainy season does not start in the same month in every country; moreover, the very south of the region (say, the coastal regions of Côte d'Ivoire, Ghana, etc.) have two distinct seasonal rain peaks, typically in June and September, with a drier lull inbetween as the ITCZ moves south -> north -> south again. Thus, speaking of "the rainy season" as if it were the same thing across the region, and basing a large part of the analysis thereon, belies the climatological differences between the West African countries/regions. This also affects the results of eg Fig 10, which changes depending on the precise definition (generic vs country-specific) of a "rainy season". I'm not saying the authors should necessarily change their analysis, but at the very least a justification for their choices is in order.

Line 67: The authors mention the mountainous areas in Nigeria, but what about the Guinée highlands where peaks >1000m are also found?

Section 2.2: I am wondering why the authors don't start with this section. After all, the satellite data are the main source for this study, with the ground-based data serving as validation material. It feels the other way around when reading this chapter, as if the ground-based data are accorded primary importance.

Line 118: "monthly mean temperature" -> why not hourly? ERA5 has much higher resolution than monthly. Is the day-night temperature effect not important for solar PV yield? Also, the authors may want to cite the paper on ERA5: https://rmets.onlinelibrary.wiley.com/doi/10.1002/qj.3803

Line 119: Here, I believe a flow chart would be highly useful, showing the different data

and modelling efforts, their characteristics, and how they feed in to the different calculations. This would include at least (i) the GHI-PV model, (ii) the validation approach for satellite data, (iii) the ERA5 data, (iv) the results (parameters), and (v) arrows indicating what feeds into what and how. This will make the paper much clearer to read and allow the reader to follow the author's train of thoughts.

Line 124: "temperature levels" -> this is explained later, but at this point in the text it's not clear what is meant with this.

Line 206: "assumed climatological AOD" -> and that assumption is what, and comes from where?

Line 248: "the wet season is actually longer in southern West Africa" -> and it is also bimodal in many places; see above comment. This is not mentioned at all in the paper.

Figure 4, 5, 6, 9: Here, I believe that the authors have placed the "Lagos" location in the wrong spot. Lagos is in south-western Nigeria, not in southern Togo.

Figure 4: I think the figure may look better if the authors used a land-sea mask. The bright colours and patterns appearing on the ocean surface are not relevant for solar PV assessments.

Line 269: Here, the authors suddenly talk about "summer months" instead of dry/wet season (but see previous comments). How are summer months defined? (I guess they refer to European summer. Is this a suitable comparison?)

Section 5.2 and 5.3: I think this order of sections is strange. I would start first with time series analysis at four locations (because this validates the use of long-term satellite data) and then explain the trend analysis afterwards. This doesn't need to be two different sections, they can be merged into one. Then, section 5.1 could be "spatial analysis" and section 5.2 "temporal analysis", or so.

Figure 6: I find the blue/red colour scheme of the "significance" figures confusing, given the similarity to the GHI graphs where the colours represent physical values instead of

a binary variable.

Figure 7 and 8: In the caption, the authors should explain what type of data is analysed here: satellite or ground-based.

Figure 10: If the authors keep their current definition of dry and wet season, perhaps it would be good to include here a vertical line showing the latitude at which, typically, the used definition (dry: October-April, wet: May-September) is the most accurate? Or else, the authors could simply replace "dry season" and "wet season" by "October-April" and "May-September" in the legend, which makes the graph fully unambigious?

Line 385: Somewhat strange that the authors here talk only about winds without even mentioning the word "clouds".

Line 389-392: Given this discussion, which is highly relevant, why don't the authors append Figure 10 with a graph of typical population density by latitude? If such data is not available, a simple solution could be to plot cities with e.g. >500,000 inhabitants as circles (radius proportional to population size) as function of latitude. This would make the point the authors try to make much more tangible.

Line 394: This reference does not seem to exist (yet). Can the authors check this?

Line 400: Why are storage capacities necessarily unavoidable to deal with dust storms? A dust storm lowers power plant availability during a few days. Power systems nowadays sometimes have to deal with power plants being unavailable during months, eg for maintenance, and yet we don't have massive storage capacities yet... Is it because dust storms are so unpredictable and massive that no reserve capacity could make up the difference? Can the authors substantiate this?

Technical corrections

Line 6: "The dry and the wet season (...)" -> suggest to delete this sentence

Line 15: Sustainable Development Goals should be capitalised

Line 17: "energies" -> "energy resources"

Line 21: "the power system needs to be built up (...)" -> suggest to write: "power systems will need to be strongly expanded in West Africa"

Line 33: "in at half hourly" -> "at half-hourly"

Line 36: I would delete the sentence comparing irradiation to a fuel.

Line 79: "as 15 min values" -> "at a 15-minute resolution"?

Line 111: "total West Africa" -> suggest to reword, perhaps "the entire region", "all continental ECOWAS countries", etc.?

Line 164: suggest to keep the names of the countries within brackets

Line 231 (and elsewhere): "temporally mean" -> "temporal mean"

Line 249: "periodes" -> "periods"

Line 251: "Mai" -> "May"

Figure 4 (and later): "interquartil" -> "interquartile"

Line 298 (and elsewhere): "tendency's" -> "tendencies"

Line 376: "dedicated" -> strange choice of words, what does this mean?

Line 381: "complementary" -> "opposite"

Line 386: "expansive" -> "expensive"

---

## Author Comment (AC1) · 10 Jun 2020

The authors would like to thank the reviewer for comments and suggestions to improve the submitted manuscript. Below, all revision points are addressed and resulting text edits are included in the following way:

- *Reviewer's points are repeated cursive.*

- Answers to the reviewer's points are given.

- "New text included to the manuscript is given in quotation marks."

[Figure]

Changed figures are all included at the end of this document.

*Title: I wonder if the long-term variability is the most important output from this paper. Isn't it rather the validated use of satellite data over the region, and the yield-latitude plots in Fig. 10? The authors may, if they agree, reconsider the suitability of the title for the paper.*
We changed the title to:
"Photovoltaic power potential in West Africa using long-term satellite data"

*Line 1: "Long-term changes" -> do the authors mean historical, or future, or both?*
We are referring to historical changes and adjusted the beginning of the sentence such that it now reads:
"This paper addresses long-term historical changes in solar irradiance [. . .]"

*Line 2: "Here we use satellite irradiance" -> and temperature from reanalysis, right?*
Yes, the sentence has been adjust in the revised version of the manuscript so that it now reads:
"Here we use satellite irradiance (Surface Solar Radiation Data Set-Heliosat, Edition 2.1, SARAH-2.1) and temperature data from a reanalysis (ERA-5) to derive photo-voltaic yields."

*Line 22: "located close to the equator, (. . .)" -> yes, but in reality, it's the locations furthest from the equator that have the highest PV potential in West Africa, as your research shows.*
We started the sentence with "West Africa" and left out the addition of where it is located.

*Line 24: "PV power system" -> this wording occurs at several instances in the paper. What exactly do the authors mean with it? Is it a power system where a certain share of power generation is from solar PV? Or solely based on PV without any other power generation sources? Is there a quantitative definition for it?*

A PV power system is a power system solely based on photovoltaic power. PV system might be a better wording for this kind of power system and is used in the new version of the manuscript.

*Line 35: "no assessment over total West Africa (. . .)" -> what is meant with "assessment"? Do the authors mean a validation of satellite data? Since this is one of the core pieces of this study, I would recommend the authors to be a lot clearer about the added value of their research here compared to the "no assessment" state-of-play.*

We changed the sentence so that it now reads:

"However, a detailed validation of the full 35 year SARAH-2.1 data set has not been performed so far for total West Africa."

*Line 42-44: "However, they need (. . .) certain assumptions." This sentence confuses me – how does it relate to the problem the authors are trying to solve? I thought the focus was long-term changes, but here it sounds as if hourly resolution is the most important problem to be solved by such research.*

The problem we tried to describe here, was that this high resolved data is often not available and that we need other solutions. Therefore, we changed this part so that it now reads:

"However, they need explicit input data in a high temporal resolution which is often not available. Therefore, a simplified model for PV yield estimations based on daily data is developed and applied here."

*Line 45: The authors do not really explain here why analysing the long-term changes*

*in West Africa is so important. Is there literature explaining why this is crucial, in particular for solar PV, either for West Africa or for other regions worldwide? Especially as compared to the variability on diurnal and seasonal timescales?*

It is important to analyze long-term data before planning and constructing a solar power plant to project the potential outcome, select the location and optimize the dimension of the power plant. We tried to make the motivation clearer in the second paragraph of the introduction (line 24-30 in the revised manuscript) and included the following text there:

"Therewith the development of a PV system is worthwhile. Before investing in a PV system three points need to be considered, using differently resolved global horizontal irradiance (GHI, the sum of direct (DIR) and diffuse horizontal irradiance (DHI)). First, to select a profitable location high spatially resolved GHI is needed. Second, to estimate the profitability and risks of the plant long-term variability and trends of historical GHI can be analyzed as a basis to project future system performance. And third, to optimize the plant high temporally resolved GHI can be used for the dimension of the plant size and storage system as well as for the maintenance. However, ground-based measurements of irradiance are not available continuously over long-term time scales and cover only a few discrete locations in the region."

*Line 59-60: I have some trouble with the definition of dry and wet season that the authors employ here – the definition seems rather generic for a region spanning a large latitude range. For example, the rainy season does not start in the same month in every country; moreover, the very south of the region (say, the coastal regions of Côte d'Ivoire, Ghana, etc.) have two distinct seasonal rain peaks, typically in June and September, with a drier lull inbetween as the ITCZ moves south -> north -> south again. Thus, speaking of "the rainy season" as if it were the same thing across the region, and basing a large part of the analysis thereon, belies the climatological differences between the West African countries/regions. This also affects the results of eg Fig 10, which changes depending on the precise definition (generic vs country-*

*specific) of a "rainy season". I'm not saying the authors should necessarily change their analysis, but at the very least a justification for their choices is in order.*
We tried to describe the difference of seasons over the entire region and why we used one single definition, when introducing the seasons:
"West Africa is a region with a pronounced dry and wet season. In large parts of West Africa one wet season occurs during the summer month. However, the length of the wet season decreases with rising latitude and along the coastal region, two wet seasons occur (typically in June/July and September). Nevertheless, here we use one single definition of seasons according to (Mohr 2004) assuming one dry season: October - April and one wet season: Mai - September. To reinforce our results we performed the analysis with a sharper definition of seasons (dry: November - March and wet: June - August) and found similar results."

*Line 67: The authors mention the mountainous areas in Nigeria, but what about the Guinée highlands where peaks >1000m are also found?*
We included all higher elevations over the entire region into the sentence so that it now reads:
"Only a few exceptions are the Mount Cameroon on the south-east of the study area along the border of Nigeria and Cameroon, Fouta Djallon and the Guinea Highlands in Guinea, Jos Plateau in the center of Nigeria and the Air Mountains in northern Niger."

*Section 2.2: I am wondering why the authors don't start with this section. After all, the satellite data are the main source for this study, with the ground-based data serving as validation material. It feels the other way around when reading this chapter, as if the ground-based data are accorded primary importance.*
We changed the order of sections (first Satellite-based data, second Ground-based data).

*Line 118: "monthly mean temperature" -> why not hourly? ERA5 has much higher resolution than monthly. Is the day-night temperature effect not important for solar PV yield? Also, the authors may want to cite the paper on ERA5:*
*https://rmets.onlinelibrary.wiley.com/doi/10.1002/qj.3803*
To provide the PV yield map shown in Fig. 10 we used daily satellite data to calculate daily PV yields. Therefore, we included daily temperatures into our model. However, for use cases, where a higher temporal resolution is required, hourly irradiance and temperature data would be appropriate. Furthermore, we included the reference for ERA5.

*Line 119: Here, I believe a flow chart would be highly useful, showing the different data and modelling efforts, their characteristics, and how they feed in to the different calculations. This would include at least (i) the GHI-PV model, (ii) the validation approach for satellite data, (iii) the ERA5 data, (iv) the results (parameters), and (v) arrows indicating what feeds into what and how. This will make the paper much clearer to read and allow the reader to follow the author's train of thoughts.*
We included a flow chart, connecting all calculation steps and needed input data and adjusted the paragraph accordingly:
"Our ultimate goal is to describe the PV potential over the entire region for a standardized PV power plant. For this purpose, a simplified linear regression is fitted on the basis of the three reference sites where the necessary information is available. Furthermore, the uncertainties concerning cell temperature are estimated (see Section 3.2) and the used GHI (from SARAH-2.1 data set) is validated (see Section 4). The single calculation steps, including all necessary input data is shown in Figure 1."

*Line 124: "temperature levels" -> this is explained later, but at this point in the text it's not clear what is meant with this.*
We deleted the sentence here and the information about the source of the temperature (from ERA5) is included later.

*Line 206: "assumed climatological AOD" -> and that assumption is what, and comes from where?*
Here, we mean the climatological AOD used for the SARAH data retrieval (see Figure 1d). We included the reference to the Figure when we first mention the climatological AOD in this paragraph:
"To study whether deviations from the climatological AOD used in SARAH-2.1 (see Figure 1 d) might explain the deviation we investigate the impact of the difference between the measured AOD and the climatological AOD for the [. . .]"

*Line 248: "the wet season is actually longer in southern West Africa" -> and it is also bimodal in many places; see above comment. This is not mentioned at all in the paper.*
See answer to your comment above.

*Figure 4, 5, 6, 9: Here, I believe that the authors have placed the "Lagos" location in the wrong spot. Lagos is in south-western Nigeria, not in southern Togo.*
You are right, thanks for this comment. We corrected the location in all Figures.

*Figure 4: I think the figure may look better if the authors used a land-sea mask. The bright colours and patterns appearing on the ocean surface are not relevant for solar PV assessments.*
We included a land-sea mask to all image plots, as only the land areas are important for solar power generation.

*Line 269: Here, the authors suddenly talk about "summer months" instead of dry/wet season (but see previous comments). How are summer months defined? (I guess they refer to European summer. Is this a suitable comparison?)*

We changed the term "summer month" to "wet season" so that it now reads:
"[. . .] occurring during the wet season."

*Section 5.2 and 5.3: I think this order of sections is strange. I would start first with time series analysis at four locations (because this validates the use of long-term satellite data) and then explain the trend analysis afterwards. This doesn't need to be two different sections, they can be merged into one. Then, section 5.1 could be "spatial analysis" and section 5.2 "temporal analysis", or so.*
We changed the order and named the sections according to your suggestions in the revised manuscript.

*Figure 6: I find the blue/red colour scheme of the "significance" figures confusing, given the similarity to the GHI graphs where the colours represent physical values instead of a binary variable.*
Figure 9 in the revised manuscript (here Figure 2): We included the information about the significance in the trend plots, so that only the significant trends are pictured in three maps (see Figure 2). This has the additional effect of reducing the size of the document.
The full caption of this figure now reads:
"Linear trend for global irradiance of the annual mean (a), as well as the dry (b) and the wet season (c), each for all significant cases. Ouagadougou, Burkina Faso and Dakar, Senegal are additionally visualized here, as values at these locations are compared within this section."

*Figure 7 and 8: In the caption, the authors should explain what type of data is analysed here: satellite or ground-based.*
We included the information on the data source in the caption.

**[ACPD](https://www.atmospheric-chemistry-and-physics.net/)**

Interactive
comment

*Figure 10: If the authors keep their current definition of dry and wet season, perhaps it would be good to include here a vertical line showing the latitude at which, typically, the used definition (dry: October-April, wet: May-September) is the most accurate? Or else, the authors could simply replace "dry season" and "wet season" by "October-April" and "May-September" in the legend, which makes the graph fully unambigious?*
Figure 11 in the revised manuscript (here Figure 3): We included the latitude range as a gray box in the background of the Figure, where the definition of seasons is the most accurate. The full caption of this figure now reads:
"Mean (temporal) PV yield at each latitude, for the total year (a), population density for each latitude (b, (NASA 2020)), as well as mean PV yield at each latitude for the dry: October-April (light grey) and wet season: May-September (dark gray) (c), in the longitude range between 4°W and 4°E. The stars in (a) mark the temporal mean PV yield calculated with the explicit model and measured ambient temperature at the three sites, Agoufou (2005-2008), Banizoumbou (2005-2012) and Djougou (2002-2009). The gray background box in (c) marks the latitude range, where the definition of seasons is most accurate."

*Line 385: Somewhat strange that the authors here talk only about winds without even mentioning the word "clouds".*
We restructured the sentence so that it now reads:
"This seasonality is dominated by the moist monsoon winds, going along with high cloudiness and coming from the south-west during the wet season and the dry Harmattan winds from the north-east during the dry season."

*Line 389-392: Given this discussion, which is highly relevant, why don't the authors append Figure 10 with a graph of typical population density by latitude? If such data is not available, a simple solution could be to plot cities with e.g. >500,000 inhabitants as circles (radius proportional to population size) as function of latitude. This would make the point the authors try to make much more tangible.*

We included a plot of population density for the corresponding longitude box in the Figure, here Figure 3 (Figure 11 in the revised manuscript), using data from the NASA (Gridded population density (NASA 2020)).

Furthermore, we included a sentence under the Figure to describe the plot:

"Population density shows the opposite latitudinal gradient compared to PV potential, with a higher density at low and a lower density at high latitudes (see Figure 11b)."

*Line 394: This reference does not seem to exist (yet). Can the authors check this?*

The reference was still in the review process, when we submitted this manuscript. In the meantime, the title changed and the manuscript was recently published in Nature Sustainability. We included the right reference in the list (Sterl 2020).

*Line 400: Why are storage capacities necessarily unavoidable to deal with dust storms? A dust storm lowers power plant availability during a few days. Power systems nowadays sometimes have to deal with power plants being unavailable during months, eg for maintenance, and yet we don't have massive storage capacities yet... Is it because dust storms are so unpredictable and massive that no reserve capacity could make up the difference? Can the authors substantiate this?*

Of course we do not need such high storage capacities if different power sources are used and reserve capacities can be used from other power sources if there is only few solar irradiance available. However, in the conclusion of this study, we describe a solely based solar system, where these storage capacities would be necessary, because no other power sources exist. By combining solar power with other power sources, storage capacities can be reduced drastically due to compensating possibilities. To make clear, that this statement is for a solely based power system, we included the word 'solely' in the sentence so that it now reads:

"For such events storage capacities for several days might be needed e.g. in solely solar based micro grids."

[Figure]

Technical corrections were included into the manuscript.

**References**

Mohr, K. I.: Interannual, monthly, and regional variability in the Wet season diurnal cycle of precipitation in sub-Saharan Africa, Journal of Climate, 2004.

NASA: Gridded population density, https://sedac.ciesin.columbia.edu/data/set/gpw-v4-population-density-rev11/data-download, 2020.

Sterl, S., Vanderkelen, I., Chawanda, C. J., Russo, D., Brecha, R. J., van Griensven, A., Van Lipzig, N. P., and Thiery, W.: Smart renewable electricity portfolios in West Africa, Nature Sustainability, 2020.

[Figure]

[Figure]

**Fig. 1.** Connection of calculation steps (red) within this study, including all needed input data (green: satellite data, gray: reanalysis data, blue: observational data).

**(a)**

**trend in GHI**

15 N Dakar

• Agoufou

• Banizoumbou

• Ouagadougou

10 N

• Djougou

• Lagos

5 N   15 W   10 W   5 W   0   5 E   10 E

(W/m²/decade)

−5         −2.5         0         2.5         5

**(b)**

**trend in GHI (dry season)**

15 N

• Agoufou

• Banizoumbou

10 N

• Djougou

• Lagos

5 N   15 W   10 W   5 W   0   5 E   10 E

(W/m²/decade)

−5         −2.5         0         2.5         5

**(c)**

**trend in GHI (wet season)**

15 N

• Agoufou

• Banizoumbou

10 N

• Djougou

• Lagos

5 N   15 W   10 W   5 W   0   5 E

(W/m²/decade)

−5         −2.5         0         2.5         5

**Fig. 2.** Linear trend for global irradiance of the annual mean (a), as well as the dry (b) and the wet season (c), each for all significant cases.

[Figure]

**Fig. 3.** Mean PV yield at each latitude, for the total year (a), population density for each latitude (b, (NASA 2020)), as well as mean PV yield for the dry (light grey) and wet season (dark gray) (c).

---

## Referee Comment (RC2) · Anonymous Referee #3 · 12 Jun 2020

General comments

In this work, I can distinguish two main parts: First, an evaluation was performed for 35-year time series of SARAH-2.1 global and direct surface horizontal irradiance against ground-based measurements from three stations in West Africa and the spatial and temporal variability of these data was assessed. Second part was the use of satellite data to derive empirically pv yields for West Africa region and these results were analyzed in space and time and were validated by explicitly calculated values from the aforementioned ground-based stations. As long as solar energy renewable sources is a promising and sustainable source of energy, this study provides interesting results

concerning PV power potential in West Africa. I would consider to recommend this paper for publication, if the authors can address the following comments:

Specific comments

Lines 22-23: you state climatological conditions governing a great area, you may support it with relevant bibliography?

Lines 64-71: Apart from the study's area topography, from which I suggest to begin with, which provides the valuable information of mountain regions essential for cloud formation, you provide climatological conditions of study area especially surface albedo, mean cloud albedo, aerosol optical depth. You provide this information by figure 1 and there you explain briefly how these values are computed. Please, describe here how those values and fig 1 produced giving information in the manuscript about the data used.

Line 118: "for monthly mean temperature" maybe you mean daily mean temperature as you mention at line 124

Line 155: GTI and not GHI?

Lines 159-160: maybe: the parameter b ... . The slope $\alpha$?

Lines 212-213: the percentages inside parenthesis are reductions of RMSE? Are the right values because it doesn't make sense for example for Afougou compared to the values given in fig. 3

Line 285: "... being significant" please rephrase that sentence and give additional information of how you assess the statistical significance of the linear trends?

Figure 8 caption: Trends of monthly mean anomalies were calculated and provided on the plots, if they were found to be statistically significant, please provide information about how you assess the statistical significance.

Figure 10 caption: The central line of those box plots provides mean value or median?

Please explain and if is the median perhaps you should provide on this figure the median of the explicitly calculated PV yields for the three sites.

Figure 10 caption: Instead of "temporally" temporal variations.

Technical corrections

Lines 8, 10: use parenthesis instead of W/m2/decade use (W/m2)/decade and everywhere applicable in the manuscript

Line 31: Organization not "Organisation"

Lines 35: and for the full 35 ... instead of "and the full 35 ..."

Line 60: May not "Mai"

Line 80: measurement not "measuremten"

Line 119: and is available ... instead of "and available"

Line 181: and decreases (0.78) [without to]

Figure 2: the symbols used to describe the regression lines must be consistent through the manuscript, so please select only one symbol for the slope

Line 186: the verb is missing in order to complete your sentence, so in order to make sense please rephrase you sentence

Figure 3: At the second sentence please remove second "the".

Line 231: . . .the temporal mean. . . instead of ". . .the temporally mean. . ."

Figure 4 caption: (1983 to 2017) . . . instead of "(1983 and 2017)"

Figure 4: Main title of c,d: interquartile

Figure 4 caption: temporal variations instead of "temporally"

Line 253: temporal variations of IQR and not "temporally IQR" and everywhere applicable in the manuscript

Figure 5: Main title of c,d: interquartile

Line 269: summer months. . . instead of "summer month"

line 318: ... time series of monthly mean ... instead of "... time series monthly mean ..."

Figure 9: Main title of b: interquartile

Line 386: . . . is expensive. instead of ". . . is expansive."

Line 389: ... the urban regions in the south. instead of "... the urban regions is the south."
* * *

---

## Referee Comment (RC3) · Anonymous Referee #1 · 17 Jun 2020

**1   General Comments**

This article looks at surface solar irradiance and photovoltaic power generation in West Africa. The authors use surface measurements of solar irradiance to evaluate satellite-based estimates of solar irradiance, which provide better temporal and spatial coverage. They then use the satellite-based estimate of solar surface irradiance to characterize its regional and temporal variability. Estimates of photovoltaic power yield are calculated as a simple linear function of the surface solar irradiance and again its spatial and temporal variability is characterized.

[Figure]

The research described in this article is original, though the results are rather predictable based on the existing literature. Scientifically, I think the analysis is fairly sound, though I have some concerns about the implications of the errors in the satellite estimates of solar surface irradiance. The quality of the writing is reasonable. I have made some suggestions for improvements below.

1. The difference between the surface and SARAH estimates of GHI are rather large at two of the sites. Assuming that these errors are representative of the uncertainty in the SARAH product across the region, how does this impact on the subsequent analysis of spatial/temporal variability and trends (the errors are comparable to the scale of much of the spatial and temporal variabilities presented in section 5 and much larger than the total trend estimates). The section concludes "the evaluation shows that the SARAH-2.1 data record can be used to get a reasonable overview on the irradiance variability and trends to estimate the PV potential in West Africa". What magnitude errors would mean that the SARAH data record isn't suitable?

2. On a related note, I would be interested to see how the errors impact on the photovoltaic power yield estimates. How different would the estimates at the three surface sites be if you used the surface measured GHI rather than the satellite GHI as input?

3. I would consider changing the structure of the paper so that the description of the methodologies to calculate photovoltaic power yield (i.e section 3) is placed after the results for GHI and immediately before the presentation of the results for the photovoltaic power yield.

**2 Specific Comments**

1. I'm not convinced the surface albedo shown in Fig 1(a) is particularly relevant for this study as it has no impact on the GHI. I'd suggest using a different image instead. Perhaps a snapshot visible image from SEVIRI?

2. The paragraph comparing MVIRI and SEVIRI (L100) seems incomplete. Which channels are used for SARAH? Which channels are on SEVIRI? When does SARAH use MVIRI and when SEVIRI?

3. Please can you specify what the MAE quoted for SARAH (L108) is measured against? Is this compared to surface-based observations? If so where and when?

4. I would move the text on ERA5 data (L117-120) to the following paragraph.

5. If I understand correctly, "b" in equation (5) represents the power required by the inverter, which is a function of temperature. Yet in table 3, for T>35 it has a positive value, which implies the inverter is generating power? Can you comment on this?

6. Why are some of the points in Figure 3 grey? Are these points where Delta AOD is negative?

7. For the trend analysis, can you take the uncertainty in the SARAH measurements into account in your estimates of the significance of the trend?

**3 Technical Corrections**

**L25**: I don't think "therewith" is the correct word here (nor in the other places where it is used in the article).

**L35**: Remove total

**L38**: "sun zenith" should be "solar zenith"

**L39**: "shadings from the surrounding" should be "shade from the surroundings"

**L98**: "ground albedo" should be "surface albedo"

**L160-L163**: Are "a" and "b" the wrong way round here?

**Fig. 2**: Change legend to b1+a1, etc for consistency with text.

**L196-199**: Pair of sentences about SARAH performance n Banizoumbou can be made more concise e.g. "In Banizoumbou, SARAH performance is consistent with previous evaluation of against BSRN stations".

**L231**: "temporally" should be "temporal"

**L249**: "periodes" should be periods

**L250**: "Mai" should be May

**L269**: "summer month" should be summer months

**L272**: change "are the most efficient modulator" to "have a large impact on"

**L333**: "providing" should be "demonstrating"
* * *

---

## Author Comment (AC2) · 14 Jul 2020

The authors would like to thank the reviewer for comments and suggestions to improve the submitted manuscript. Below, all revision points are addressed and resulting text edits are included in the following way:

- *Reviewer's points are repeated cursive.*

- Answers to the reviewer's points are given.

- "New text included to the manuscript is given in quotation marks."

[Figure]

*Lines 22-23: you state climatological conditions governing a great area, you may support it with relevant bibliography?*

We revised the text accordingly:

"West Africa receives high amounts of global horizontal irradiance (GHI) (Solargis 2019). With the descending branch of the Hadley Cell the Sahara and the Sahel zone are overall dry with little cloudiness leading to high sunshine duration (Kothe 2017)."

*Lines 64-71: Apart from the study's area topography, from which I suggest to begin with, which provides the valuable information of mountain regions essential for cloud formation, you provide climatological conditions of study area especially surface albedo, mean cloud albedo, aerosol optical depth. You provide this information by figure 1 and there you explain briefly how these values are computed. Please, describe here how those values and fig 1 produced giving information in the manuscript about the data used.*

We changed the order in Figure 1 to a) topography, b) cloud albedo, c) surface albedo and d) aerosol optical depth and included the source of the data in the text, as it now reads:

"West Africa is in general rather flat with highest elevations typically below 1000 m (Figure 1 a, Global Land One-km Base Elevation Project (GLOBE) database (Hastings 1999)). Some exceptions are the Mount Cameroon on the south-east of the study area along the border of Nigeria and Cameroon, Fouta Djallon and the Guinea Highlands in Guinea, Jos Plateau in the center of Nigeria and the Aïr Mountains in northern Niger. Here, but locally also for lower mountain ranges, orographically enhanced cloudiness might occur. The enhanced cloudiness associated to the moist tropical region is clearly visible in the mean cloud albedo used as input for the SARAH-2.1 data retrieval between 1983 and 2017 (see Figure 1 b, from the SARHA-2.1 data set described later). Clouds have the major influence on the irradiance analyzed in this study. The West African climate zones relate to the albedo climatology (used for the SARAH-2.1 data retrieval), with a higher albedo of up to 0.35 in the desert region in the north and

a lower albedo of down to 0.1 in the forest region in the south (see Figure 1 c, Surface and Atmospheric Radiation Budget (SARB) data from Clouds and the Earth's Radiant Energy System (CERES)). Frequent dust outbreaks occur over the total region (Cowie 2014). Thereby, climatological highest aerosol optical depth (AOD) of up to 0.35 can be found in northern Mali (see Figure 1 d, from the European Center for Medium Range Weather Forecast, Monitoring Atmospheric Composition and Climate (MACC) and used for the SARAH-2.1 data retrieval). "

Besides clouds, aerosols can have a significant impact on the analyzed irradiance. We added a sentence at the end of the paragraph:

"Besides clouds, aerosols can have a high impact on the irradiance and therewith on solar power (Neher 2019)."

*Line 118: "for monthly mean temperature" maybe you mean daily mean temperature as you mention at line 124*
Yes, you are right. We changed this in the revised manuscript.

*Line 155: GTI and not GHI?*
We left out this part of the sentence, as it might be confusing at this point which data is used later. However, we explain later, which data is used for the calculations.

*Lines 159-160: maybe: the parameter b ... . The slope $\alpha$?*
Yes, we interchanged the parameters and changed the sentence, as it now reads:
"The parameter $b$ indicates the impact of the inverter, as it needs a certain amount of power to work. The slope $a$ indicates the efficiency, including the conversion of W/m$^2$ to kWh/kWp."

*Lines 212-213: the percentages inside parenthesis are reductions of RMSE? Are the*

*right values because it doesn't make sense for example for Afougou compared to the
values given in fig. 3*
The values inside the parenthesis are reductions of RMSE. The RMSE given in Fig. 3
is reduced by these values, if only the situations with AOD<0.05 are used.

*Line 285: "... being significant" please rephrase that sentence and give additional
information of how you assess the statistical significance of the linear trends?*
How the statistical significance is assessed is given in lines 234-236 (revised
manuscript: 243-245):
"The significance of the trend is checked by calculating the 95% confidence interval.
The trends are significantly positive (negative) if the upper and lower value of the 95%
confidence interval are positive (negative)."
However, we included a short definition of significance here again:
"However, the absolute values of the trend reach around $\pm 5$ W/m$^2$/decade and being
significant (based on the 95% confidence interval)."

*Figure 8 caption: Trends of monthly mean anomalies were calculated and provided on
the plots, if they were found to be statistically significant, please provide information
about how you assess the statistical significance.*
Due to the comment of another referee we changed the figure (Figure 9 in revised
manuscript). Now only statistically significant cases are shown for the trend. We
included the definition of significance again in the caption of the figure:
"Figure 9. Linear trend for global irradiance of the annual mean (a), as well as the dry
(b) and the wet season (c), each for all significant cases (based on the 95% confidence
interval). Ouagadougou, Burkina Faso and Dakar, Senegal are additionally visualized
here, as values at these locations are compared within this section."

*Figure 10 caption: The central line of those box plots provides mean value or median?*

*Please explain and if is the median perhaps you should provide on this figure the median of the explicitly calculated PV yields for the three sites.*
The central line of the box plots provides the median of the regional distribution within each latitude. The PV yield, however, is given as the temporal mean. Therefore, we also provided the temporal mean at the single locations, as these do not have a regional distribution. Furthermore, we included a sentence on what is shown in the figure, just before the figure:
"Figure 10 (Figure 11 in revised manuscript) shows the variability of the temporal mean PV yield for each latitude separately.".

*Figure 10 caption: Instead of "temporally" temporal variations.*
We changed the word temporally to temporal in the caption of Figure 10.

Technical corrections were included into the manuscript.

**References**

Kothe, S., Pfeifroth, U., Cremer, R., Trentmann, J., and Hollmann, R.: A satellite-based sunshine duration climate data record for Europe and Africa, Remote Sensing, 9, 429, 2017.

Neher, I., Buchmann, T., Crewell, S., Pospichal, B., and Meilinger, S.: Impact of atmospheric aerosols on solar power, Meteorologische Zeitschrift, 28, 305–321, 2019.

Solargis: Solar Resource Map 2019, https://solargis.com/maps-and-gis-data/download/africa, 2019.

---

## Author Response (AR1)

**Response to comment from Referee #1 on "Long-term variability of solar irradiance and its implications for photovoltaic power in West Africa" by Ina Neher et al.**

Ina Neher, ina.neher@h-brs.de and co-authors

July 16, 2020

The authors would like to thank the reviewer for comments and suggestions to improve the submitted manuscript. Below, all revision points are addressed and resulting text edits are included in the following way:

- *Reviewer's points are repeated cursive.*

- Answers to the reviewer's points are given.

- "New text included to the manuscript is given in quotation marks."

As the revised manuscript was slightly changed in some formulations after the public discussion, we give you the answers similar to the answer in the public discussion, but with all line numbers and the exact text of the uploaded version of the revised manuscript.

**General comments**

*The difference between the surface and SARAH estimates of GHI are rather large at two of the sites. Assuming that these errors are representative of the uncertainty in the SARAH product across the region, how does this impact on the subsequent analysis of spatial/temporal variability and trends (the errors are comparable to the scale of much of the spatial and temporal variabilities presented in section 5 and much larger than the total trend estimates). The section concludes "the evaluation shows that the SARAH-2.1 data record can be used to get a reasonable overview on the irradiance variability and trends to estimate the PV potential in West Africa". What magnitude errors would mean that the SARAH data record isn't suitable?*
The lack in data availability over the entire region (less than 20% of the time period of satellite data and only three sites) makes it difficult to generally validate the satellite product. In general, we find a high correlation between SARAH-2.1 estimates and GHI observations at all sites. The RMSE and MAE are given as non bias-corrected values. The bias, which dominates the RMSE and MAE, lies in the range of the uncertainties of ground based measurements (2% for Banizoumbou and Djougou and 10% for Agoufou) in Banizoumbou and Agoufou. At Djougou we find an offset of around 12%, which was mentioned and discussed in the manuscript and has been reported in other studies (Hannak et al. 2017). The offset in this region is known and would even strengthen our

results, as an overestimated GHI in southern West Africa increases the actual north-south gradient of surface irradiance. Given the high correlation and a total uncertainty being lower than the variability of solar irradiance in the region, we judge the satellite data being reasonable enough to show general differences in PV yields over the entire region. We expanded the discussion in Section 4 (line 240-247), as it now reads:

"Given the good correlation and the fact that the uncertainty is dominated by the bias the evaluation supports the suitability of the data set to investigate the variability of solar irradiance. Thus, the SARAH-2.1 data record can be used to get an overview on the irradiance variability and trends to estimate the PV potential in West Africa. However, especially in southern West Africa the systematical overestimation of solar irradiance in the SARAH-2.1 data set (Kniffka et al. 2019, Hannak et al. 2017) need to be considered in the conclusions of the variability and trend analysis. As a consequence of the positive offset in southern West Africa, the actual north-south gradient in the satellite data set is underestimated. In particular, for the trend analysis the systematic offset would not have an impact. Overall, an expansion of measurements over longer time periods (the measured data is available for less than 20% of the time period at only three sites) could increase the significance of our validation."

*On a related note, I would be interested to see how the errors impact on the photovoltaic power yield estimates. How different would the estimates at the three surface sites be if you used the surface measured GHI rather than the satellite GHI as input?*

As we used a linear PV yield model, the uncertainty in GHI would propagate linearly. We include a sentence into the revised manuscript at the beginning of Section 6 (line 349-350):

"As we used a linear approach, the uncertainty of satellite data would propagate linearly for PV yield estimates."

As the explicit PV yield model we used for the development of the simple PV yield model needs DNI besides GHI as an input, we could only model PV yields by using the measured GHI with the simplified model. For illustration, we include the points calculated with the simple model and measured GHI at the three location in the last figure (Figure 11 in revised manuscript). Furthermore, we include a short discussion on the results after the figure, reading as (line 365-370):

"In general, the overestimation of satellite data in Agoufou and Djougou as well as the slight underestimation in Banizoumbou (see Section 4) can be seen in the PV yields calculated with the simple model and using the measured GHI as an input (crosses in Figure 11 a). In Agoufou, the PV yields, calculated with the linear model, are similar to the explicitly calculated PV yields. In Banizoumbou the results are higher and in Djougou they are lower compared to the PV yields calculated with satellite data. Especially in Djougou, the irradiance decreases over the 35 years of satellite data availability. This leads to lower values in the 2000's values compared to the mean."

*I would consider changing the structure of the paper so that the description of the methodologies to calculate photovoltaic power yield (i.e section 3) is placed after the results for GHI and immediately before the presentation of the results for the photovoltaic power yield.*

For the consistence of the story line, we decided to better describe the methodology first and the results thereafter. Furthermore, some parts of the structure were changed according to other reviewers comments.

**Specific comments**

[Figure]

Figure 11: Mean (temporal) PV yield at each latitude, for the total year (a), population density for each latitude (b, (NASA 2020)), as well as mean PV yield at each latitude for the dry: October-April (light grey) and wet season: May-September (dark gray) (c), in the longitude range between 4°W and 4°E. The single points mark the temporal mean PV yield calculated with the explicit model and measured ambient temperature (star) as well as the PV yield calculated with the simple model and measured GHI (cross) at the three sites, Agoufou (2005-2008), Banizoumbou (2005-2012) and Djougou (2002-2009). The gray background box in (c) marks the latitude range, where the definition of seasons is most accurate.

*I'm not convinced the surface albedo shown in Fig 1(a) is particularly relevant for this study as it has no impact on the GHI. I'd suggest using a different image instead. Perhaps a snapshot visible image from SEVIRI?*

When GHI is retrieved from satellite reflectance the albedo is an important input parameter. In Figure 1 of the manuscript we show (besides the topography) the input data for the SARAH-2.1 data retrieval. In the revised manuscript we changed the order of the figures, to the topography being a). Furthermore, we added a phrase to the text, that the other data is used as input for the SARAH-2.1 data retrieval. For completeness we kept the albedo figure, as the albedo impacts the diffuse part of GHI.

*The paragraph comparing MVIRI and SEVIRI (L100) seems incomplete. Which channels are used for SARAH? Which channels are on SEVIRI? When does SARAH use MVIRI and when SE-VIRI?*

We tried to be clearer in our formulation and changed the paragraph to as it now reads (line 98-101): "For the generation of the SARAH-2.1 data record the visible channel (0.5 - 0.9 $\mu m$) of the ME-TEOSAT Visible and Infrared Imager (MVIRI) is used until 2005 and the two visible channels (0.6 and 0.8 $\mu m$) of the Spinning Enhanced Visible and Infrared Imager (SEVIRI) afterward. A detailed description of the retrieval is given in (Mueller 2015) and references within."

*Please can you specify what the MAE quoted for SARAH (L108) is measured against? Is this compared to surface-based observations? If so where and when?*
The SARAH data was compared to ground-based measurements at 15 BSRN stations. The earliest measurements started in the mid 1990's. We specified the measurement to which the SARAH data was compared in the sentence, so that it now reads (line 105-107):
"A mean absolute error (MAE, in comparison to 15 BSRN stations between 1994 and 2017) of 5.5 W/m$^2$ and 11.7 W/m$^2$ for monthly and daily GHI is reached, respectively (Pfeifroth 2019)."

*I would move the text on ERA5 data (L117-120) to the following paragraph.*
Thank you for your suggestion, we moved the description of the ERA5 data to the section on PV yield estimations (Section 3).

*If I understand correctly, "b" in equation (5) represents the power required by the inverter, which is a function of temperature. Yet in table 3, for T>35 it has a positive value, which implies the inverter is generating power? Can you comment on this?*
The parameter "b" describes the additional impact of the inverter for PV power estimations. The inverter needs a certain amount of solar irradiance to convert the direct current to alternating current. But you are right, actually it should not have a positive value for physical reasons. This effect occurs because at high temperatures, irradiance is comparably high. We corrected the parameter b for T>35°C to zero and included a description in the revised manuscript. Furthermore, we repeated our calculations with the new parameters (line 196-197).
"The slope $a$ decreases at increasing temperatures. For T>35°C the parameter $b$ was set to zero, as for physical reasons it can not be positive."

*Why are some of the points in Figure 3 grey? Are these points where Delta AOD is negative?*
If there was no AOD measurements from AERONET available, we did not mark the points in color. We included a remark on this in the caption of the figure (Figure 4 in revised manuscript), as the full caption now reads as:
"Comparison of simulated and observed GHI as daily (left) and monthly (right) averages at three sites over the given timely horizon, a) Agoufou (2005-2008), b) Banizoumbou (2005-2012) and c) Djougou (2002-2009). The difference between the measured AOD and the climatological AOD for the satellite data retrieval ($\Delta$AOD) is indicated as color. If no measured AOD is available, the points are grey."

*For the trend analysis, can you take the uncertainty in the SARAH measurements into account in your estimates of the significance of the trend?*
We tried to include the uncertainties into the discussion of the results for the trend analysis and changed the manuscript accordingly (line 318-322).
"Compared to the uncertainties of the satellite data (MAE up to 27.6 W/m$^2$, see Section 4), the trends might seem negligible. However, the reported uncertainties are not bias corrected and represent, in particular in the case of Djougou, the systematic overestimation of the GHI by the satellite estimate. The estimation of the temporal trend is unaffected by any systematic over- or underestimation and, hence, still can be derived with certain confidence."

Technical corrections were included into the manuscript.

**References**

Hannak, L., Knippertz, P., Fink, A. H., Kniffka, A., and Pante, G.: Why do global climate models struggle to represent low-level clouds in the west african summer monsoon?, Journal of Climate, 30, 1665–1687, 2017.

Kniffka, A., Knippertz, P., and Fink, A. H.: The role of low-level clouds in the West African monsoon system, Atmospheric Chemistry and Physics, 19, 1623–1647, 2019.

Mueller, R., Pfeifroth, U., Traeger-Chatterjee, C., Trentmann, J., and Cremer, R.: Digging the METEOSAT Treasure—3 Decades of Solar Surface Radiation, Remote Sensing, 7, 8067–8101, 2015.

NASA: Gridded population density, https://sedac.ciesin.columbia.edu/data/set/gpw-v4-population-density-rev11/data-download, 2020.

Pfeifroth, U., Trentmann, J., and Kothe, S.: Validation Report: Meteosat Solar Surface Radiation and Effective Cloud Albedo Climate Data Record SARAH-2 . 1 climate data records, Tech. rep., DWD, 2019.

**Response to comment from Referee #2 on "Long-term variability of solar irradiance and its implications for photovoltaic power in West Africa" by Ina Neher et al.**

Ina Neher, ina.neher@h-brs.de and co-authors

July 16, 2020

The authors would like to thank the reviewer for comments and suggestions to improve the submitted manuscript. Below, all revision points are addressed and resulting text edits are included in the following way:

- *Reviewer's points are repeated cursive.*

- Answers to the reviewer's points are given.

- "New text included to the manuscript is given in quotation marks."

As the revised manuscript was slightly changed in some formulations after the public discussion and due to other reviewers comments, we give you the answers similar to the answer in the public discussion, but with all line numbers and the exact text of the uploaded version of the revised manuscript.

*Title: I wonder if the long-term variability is the most important output from this paper. Isn't it rather the validated use of satellite data over the region, and the yield-latitude plots in Fig. 10? The authors may, if they agree, reconsider the suitability of the title for the paper.*
We changed the title to:
"Photovoltaic power potential in West Africa using long-term satellite data"

*Line 1: "Long-term changes" -> do the authors mean historical, or future, or both?*
We are referring to historical changes and adjusted the beginning of the sentence such that it now reads (line 2):
"This paper addresses long-term historical changes in solar irradiance [. . .]"

*Line 2: "Here we use satellite irradiance" -> and temperature from reanalysis, right?*
Yes, the sentence has been adjust in the revised version of the manuscript so that it now reads (line 3-4):
"Here we use satellite irradiance (Surface Solar Radiation Data Set-Heliosat, Edition 2.1, SARAH-2.1) and temperature data from a reanalysis (ERA-5) to derive photovoltaic yields."

*Line 22: "located close to the equator, (. . .)" -> yes, but in reality, it's the locations furthest from the equator that have the highest PV potential in West Africa, as your research shows.*

We changes the sentence to (line 21):
"With regard to energy availability and security West Africa is one of the least developed regions in the world (ECOWAS 2017)."

*Line 24: "PV power system" -> this wording occurs at several instances in the paper. What exactly do the authors mean with it? Is it a power system where a certain share of power generation is from solar PV? Or solely based on PV without any other power generation sources? Is there a quantitative definition for it?*

A PV power system is a power system solely based on photovoltaic power. PV system might be a better wording for this kind of power system and is used in the new version of the manuscript.

*Line 35: "no assessment over total West Africa (. . .)" -> what is meant with "assessment"? Do the authors mean a validation of satellite data? Since this is one of the core pieces of this study, I would recommend the authors to be a lot clearer about the added value of their research here compared to the "no assessment" state-of-play.*

We changed the sentence so that it now reads (line 43):
"However, a detailed validation of the full 35 year SARAH-2.1 data set has not been performed so far for total West Africa."

*Line 42-44: "However, they need (. . .) certain assumptions." This sentence confuses me – how does it relate to the problem the authors are trying to solve? I thought the focus was long-term changes, but here it sounds as if hourly resolution is the most important problem to be solved by such research.*

The problem we tried to describe here, was that this high resolved data is often not available and that we need other solutions. Therefore, we changed this part so that it now reads (line 49-51):
"However, they need explicit input data in a high temporal resolution which is often not available. Therefore, a simplified model for PV yield estimations based on daily data is developed and applied here."

*Line 45: The authors do not really explain here why analysing the long-term changes in West Africa is so important. Is there literature explaining why this is crucial, in particular for solar PV, either for West Africa or for other regions worldwide? Especially as compared to the variability on diurnal and seasonal timescales?*

It is important to analyze long-term data before planning and constructing a solar power plant to project the potential outcome, select the location and optimize the dimension of the power plant. We tried to make the motivation clearer in the second paragraph of the introduction and included the following text there (line 26-32):
"Thus, the development of a PV system is worthwhile. Before investing in a PV system three points need to be considered, using differently resolved global horizontal irradiance (GHI, the sum of direct (DIR) and diffuse horizontal irradiance (DHI)). First, to select a profitable location high spatially resolved GHI is needed. Second, to estimate the profitability and risks of the plant long-term variability and trends of historical GHI can be analyzed as a basis to project future system performance. And third, to optimize the plant high temporally resolved GHI can be used for the dimension of the plant size and storage system as well as for the maintenance. However, ground-based measurements of irradiance are not available continuously over long-term time scales and cover only a few discrete locations in the region."

*Line 59-60: I have some trouble with the definition of dry and wet season that the authors employ here – the definition seems rather generic for a region spanning a large latitude range. For example, the rainy season does not start in the same month in every country; moreover, the very south of the region (say, the coastal regions of Côte d'Ivoire, Ghana, etc.) have two distinct seasonal rain peaks, typically in June and September, with a drier lull inbetween as the ITCZ moves south -> north -> south again. Thus, speaking of "the rainy season" as if it were the same thing across the region, and basing a large part of the analysis thereon, belies the climatological differences between the West African countries/regions. This also affects the results of eg Fig 10, which changes depending on the precise definition (generic vs country-specific) of a "rainy season". I'm not saying the authors should necessarily change their analysis, but at the very least a justification for their choices is in order.*

We tried to describe the difference of seasons over the entire region and why we used one single definition, when introducing the seasons (line 64-71):

"West Africa (in this study defined as the region from 3°N to 20°N and 20°W to 16°E) is a region with a pronounced dry and wet season. In large parts of West Africa one wet season occurs during the summer months. However, the length of the wet season decreases with rising latitude and along the coastal region, two wet seasons occur (typically in June/July and September). Nevertheless, here we use one single definition of seasons according to (Mohr 2004) assuming one dry season: October - April and one wet season: May - September. To reinforce our results we performed the analysis with a sharper definition of seasons (dry: November - March and wet: June to August) and found similar results."

*Line 67: The authors mention the mountainous areas in Nigeria, but what about the Guinée highlands where peaks >1000m are also found?*

We included all higher elevations over the entire region into the sentence so that it now reads (line 75-77):

"Some exceptions are the Mount Cameroon on the south-east of the study area along the border of Nigeria and Cameroon, Fouta Djallon and the Guinea Highlands in Guinea, Jos Plateau in the center of Nigeria and the Air Mountains in northern Niger."

*Section 2.2: I am wondering why the authors don't start with this section. After all, the satellite data are the main source for this study, with the ground-based data serving as validation material. It feels the other way around when reading this chapter, as if the ground-based data are accorded primary importance.*

We changed the order of sections (first Satellite-based data, second Ground-based data).

*Line 118: "monthly mean temperature" -> why not hourly? ERA5 has much higher resolution than monthly. Is the day-night temperature effect not important for solar PV yield? Also, the authors may want to cite the paper on ERA5:*
*https://rmets.onlinelibrary.wiley.com/doi/10.1002/qj.3803*

To provide the PV yield map shown in Figure 10 (Figure 11 in the revised manuscript) we used daily satellite data to calculate daily PV yields. Therefore, we included daily temperatures into our model. However, for use cases, where a higher temporal resolution is required, hourly irradiance and temperature data would be appropriate. Furthermore, we included the reference for ERA5.

*Line 119: Here, I believe a flow chart would be highly useful, showing the different data and*

*modelling efforts, their characteristics, and how they feed in to the different calculations. This would include at least (i) the GHI-PV model, (ii) the validation approach for satellite data, (iii) the ERA5 data, (iv) the results (parameters), and (v) arrows indicating what feeds into what and how. This will make the paper much clearer to read and allow the reader to follow the author's train of thoughts.*

We included a flow chart (Figure 2 in revised manuscript), connecting all calculation steps and needed input data and adjusted the paragraph accordingly (line 131-136):

"Our ultimate goal is to describe the PV potential over the entire region for a standardized PV power plant. For this purpose, a simplified linear regression is fitted on the basis of the three reference sites where the necessary information is available. Furthermore, the uncertainties concerning cell temperature are estimated (see Section 3.2) and the used GHI (from SARAH-2.1 data set) is validated (see Section 4). Therefore, ERA5 data is used (Hersbach et al. 2020, ERA5 2017) for daily mean temperature. The ERA5 archive is based on a global reanalysis and is available from 1979 on. The single calculation steps, including all necessary input data is shown in Figure 2."

[Figure]

Figure 2: Connection of calculation steps (red) within this study, including all needed input data (green: satellite data, gray: reanalysis data, blue: observational data).

*Line 124: "temperature levels" -> this is explained later, but at this point in the text it's not clear what is meant with this.*

We deleted the sentence here and the information about the source of the temperature (from ERA5) is included later.

*Line 206: "assumed climatological AOD" -> and that assumption is what, and comes from where?*

Here, we mean the climatological AOD used for the SARAH data retrieval (see Figure 1d). We included the reference to the Figure when we first mention the climatological AOD in this paragraph (line 218-219):

"To study whether deviations from the climatological AOD used in SARAH-2.1 (see Figure 1 d) might explain the deviation we investigate the impact of the difference between the measured AOD and the climatological AOD for the [...]"

*Line 248: "the wet season is actually longer in southern West Africa" -> and it is also bimodal in many places; see above comment. This is not mentioned at all in the paper.*

See answer to your comment above.

*Figure 4, 5, 6, 9: Here, I believe that the authors have placed the "Lagos" location in the wrong spot. Lagos is in south-western Nigeria, not in southern Togo.*

You are right, thanks for this comment. We corrected the location in all Figures.

*Figure 4: I think the figure may look better if the authors used a land-sea mask. The bright colours and patterns appearing on the ocean surface are not relevant for solar PV assessments.*
We included a land-sea mask to all image plots, as only the land areas are important for solar power generation.

*Line 269: Here, the authors suddenly talk about "summer months" instead of dry/wet season (but see previous comments). How are summer months defined? (I guess they refer to European summer. Is this a suitable comparison?)*
We changed the term "summer month" to "wet season" so that it now reads (line 279-280):
"[...] occur during the wet season."

*Section 5.2 and 5.3: I think this order of sections is strange. I would start first with time series analysis at four locations (because this validates the use of long-term satellite data) and then explain the trend analysis afterwards. This doesn't need to be two different sections, they can be merged into one. Then, section 5.1 could be "spatial analysis" and section 5.2 "temporal analysis", or so.*
We changed the order and named the sections according to your suggestions in the revised manuscript.

*Figure 6: I find the blue/red colour scheme of the "significance" figures confusing, given the similarity to the GHI graphs where the colours represent physical values instead of a binary variable.*
Figure 9 in the revised manuscript: We included the information about the significance in the trend plots, so that only the significant trends are pictured in three maps (see Figure 9). This has the additional effect of reducing the size of the document.

*Figure 7 and 8: In the caption, the authors should explain what type of data is analysed here: satellite or ground-based.*
We included the information on the data source in the caption.

*Figure 10: If the authors keep their current definition of dry and wet season, perhaps it would be good to include here a vertical line showing the latitude at which, typically, the used definition (dry: October-April, wet: May-September) is the most accurate? Or else, the authors could simply replace "dry season" and "wet season" by "October-April" and "May-September" in the legend, which makes the graph fully unambigious?*
Figure 11 in the revised manuscript: We included the latitude range as a gray box in the background of the Figure, where the definition of seasons is the most accurate.

*Line 385: Somewhat strange that the authors here talk only about winds without even mentioning the word "clouds".*
We restructured the sentence so that it now reads (line 415-417):
"This seasonality is dominated by the moist monsoon winds, going along with high cloudiness and coming from the south-west during the wet season and the dry Harmattan winds from the northeast during the dry season."

*Line 389-392: Given this discussion, which is highly relevant, why don't the authors append Figure 10 with a graph of typical population density by latitude? If such data is not available, a*

**(a)**

[Figure]

Figure 9: Linear trend for global irradiance of the annual mean (a), as well as the dry (b) and the wet season (c), each for all significant cases (based on the 95% confidence interval). Ouagadougou, Burkina Faso and Dakar, Senegal are additionally visualized here, as values at these locations are compared within this section.

*simple solution could be to plot cities with e.g. >500,000 inhabitants as circles (radius proportional to population size) as function of latitude. This would make the point the authors try to make much more tangible.*

We included a plot of population density for the corresponding longitude box in the Figure, now Figure 11 in revised manuscript, using data from the NASA (Gridded population density (NASA 2020)).

Furthermore, we included a sentence under Figure 11 to describe the plot (line 373-375):

"Population density shows the opposite latitudinal gradient compared to PV potential, with a higher density at low and a lower density at high latitudes (see Figure 11b)."

*Line 394: This reference does not seem to exist (yet). Can the authors check this?*

The reference was still in the review process, when we submitted this manuscript. In the meantime, the title changed and the manuscript was recently published in Nature Sustainability. We included the right reference in the list (Sterl 2020).

*Line 400: Why are storage capacities necessarily unavoidable to deal with dust storms? A dust storm lowers power plant availability during a few days. Power systems nowadays sometimes have to deal with power plants being unavailable during months, eg for maintenance, and yet we don't have massive storage capacities yet... Is it because dust storms are so unpredictable and massive that no reserve capacity could make up the difference? Can the authors substantiate this?*

[Figure]

Figure 11: Mean (temporal) PV yield at each latitude, for the total year (a), population density for each latitude (b, (NASA 2020)), as well as mean PV yield at each latitude for the dry: October-April (light grey) and wet season: May-September (dark gray) (c), in the longitude range between 4°W and 4°E. The single points mark the temporal mean PV yield calculated with the explicit model and measured ambient temperature (star) as well as the PV yield calculated with the simple model and measured GHI (cross) at the three sites, Agoufou (2005-2008), Banizoumbou (2005-2012) and Djougou (2002-2009). The gray background box in (c) marks the latitude range, where the definition of seasons is most accurate.

Of course we do not need such high storage capacities if different power sources are used and reserve capacities can be used from other power sources if there is only few solar irradiance available. However, in the conclusion of this study, we describe a solely based solar system, where these storage capacities would be necessary, because no other power sources exist. By combining solar power with other power sources, storage capacities can be reduced drastically due to compensating possibilities. To make clear, that this statement is for a solely based power system, we included the word 'solely' in the sentence so that it now reads (line 431-432):
"For such events storage capacities for several days might be needed e.g. in solely solar based micro grids."

Technical corrections were included into the manuscript.

We changed the order in Figure 1 to a) topography, b) cloud albedo, c) surface albedo and d) aerosol optical depth and included the source of the data in the text, as it now reads (line 73-86):
"West Africa is in general rather flat with highest elevations typically below 1000 m (Figure 1 a, Global Land One-km Base Elevation Project (GLOBE) database (Hastings 1999)). Some exceptions are the Mount Cameroon on the south-east of the study area along the border of Nigeria and Cameroon, Fouta Djallon and the Guinea Highlands in Guinea, Jos Plateau in the center of

Nigeria and the Aïr Mountains in northern Niger. Here, but locally also for lower mountain ranges, orographically enhanced cloudiness might occur. The enhanced cloudiness associated to the moist tropical region is clearly visible in the mean cloud albedo used as input for the SARAH-2.1 data retrieval between 1983 and 2017 (see Figure 1 b, from the SARHA-2.1 data set described later). Clouds have the major influence on the irradiance analyzed in this study. The West African climate zones related to the albedo climatology (used for the SARAH-2.1 data retrieval), with a higher albedo of up to 0.35 in the desert region in the north and a lower albedo of down to 0.1 in the forest region in the south (see Figure 1 c, Surface and Atmospheric Radiation Budget (SARB) data from Clouds and the Earth's Radiant Energy System (CERES)). Frequent dust outbreaks occur over the total region (Cowie 2014). Thereby, climatological highest aerosol optical depth (AOD) of up to 0.35 can be found in northern Mali (see Figure 1 d, from the European Center for Medium Range Weather Forecast, Monitoring Atmospheric Composition and Climate (MACC) and used for the SARAH-2.1 data retrieval)."

Besides clouds, aerosols can have a significant impact on the analyzed irradiance. We added a sentence at the end of the paragraph (line 87-88):

"Therewith, aerosols can have a high impact on the irradiance besides clouds and thus on solar power (Neher 2019)."

*Line 118: "for monthly mean temperature" maybe you mean daily mean temperature as you mention at line 124*

Yes, you are right. We changed this in the revised manuscript.

*Line 155: GTI and not GHI?*

We left out this part of the sentence, as it might be confusing at this point which data is used later. However, we explain later, which data is used for the calculations.

*Lines 159-160: maybe: the parameter b ... . The slope α?*

Yes you are right, we interchanged the parameters and changed the sentence, as it now reads (line 168-170):

"The parameter $b$ indicates the impact of the inverter, as it needs a certain amount of power to work. The slope $a$ indicates the efficiency, including the conversion of $W/m^2$ to kilowatt hours per kilowatt-peak (kWh/kWp)."

*Lines 212-213: the percentages inside parenthesis are reductions of RMSE? Are the right values because it doesn't make sense for example for Afougou compared to the values given in fig. 3*

The values inside the parenthesis are reductions of RMSE. The RMSE given in Figure 3 (Figure 4 in revised manuscript) is reduced by these values, if only the situations with AOD<0.05 are used.

*Line 285: "... being significant" please rephrase that sentence and give additional information of how you assess the statistical significance of the linear trends?*

How the statistical significance is assessed is given in lines 252-254 of the revised manuscript:

"The significance of the trend is checked by calculating the 95% confidence interval. The trends are significantly positive (negative) if the upper and lower value of the 95% confidence interval are positive (negative)."

However, we included a short definition of significance here again (line 317-318):

"However, the absolute values of the trend reach around $\pm 5$ $W/m^2$/decade and being significant

(based on the 95% confidence interval)."

*Figure 8 caption: Trends of monthly mean anomalies were calculated and provided on the plots, if they were found to be statistically significant, please provide information about how you assess the statistical significance.*
Due to the comment of another referee we changed the figure (Figure 9 in revised manuscript). Now only statistically significant cases are shown for the trend. We included the definition of significance again in the caption of Figure 9 in the revised manuscript:
"Linear trend for global irradiance of the annual mean (a), as well as the dry (b) and the wet season (c), each for all significant cases (based on the 95% confidence interval). Ouagadougou, Burkina Faso and Dakar, Senegal are additionally visualized here, as values at these locations are compared within this section."

*Figure 10 caption: The central line of those box plots provides mean value or median? Please explain and if is the median perhaps you should provide on this figure the median of the explicitly calculated PV yields for the three sites.*
The central line of the box plots provides the median of the regional distribution within each latitude. The PV yield, however, is given as the temporal mean. Therefore, we also provided the temporal mean at the single locations, as these do not have a regional distribution. Furthermore, we included a sentence on what is shown in the figure, just before the figure (Figure 11 in the revised manuscript, line 354-355):
"Figure 11 shows the variability of the temporal mean PV yield for each latitude separately.".

*Figure 10 caption: Instead of "temporally" temporal variations.*
We changed the word temporally to temporal in the caption of Figure 10.

Technical corrections were included into the manuscript.

Ina Neher, ina.neher@h-brs.de

July 16, 2020

- The title was changed to "Photovoltaic power potential in West Africa using long-term satellite data" based on the comment from referee #2.

- We reworked on the introduction based on the referees comments.

- A clearer description of seasonality in the region is included at the beginning of section 2.

- Section 2.1 and Section 2.2 was exchanged, based on the comment from referee #2.

- A new figure was included (Figure 2), giving an overview on the single calculation steps (based on the comment from referee #2).

- The accuracy of the SARAH-2.1 data set was discussed in more detail, based on the comment from referee #1.

- The results for the variability and trend analysis of solar irradiance are now described in two subsections (5.1 - Spacial analysis and 5.2 - Temporal analysis), based on the comment from referee #2.

- The impact of the accuracy of the SARAH-2.1 data set on the trend analysis and PV yield calculations is now described and discussed, based on the comment from referee #1.

- Figure 11 was expanded by including a graph on the population density over latitudes, based on the comment from referee #2. Furthermore, the calculated PV yields with the simple model and the measured GHI are included for the three sites, based on the comment from referee #1.

[revised manuscript text omitted]